# Inference for Optimal Linear Treatment Regimes in Personalized Decision-making

**Yuwen Cheng**[1]

**Shu Yang**[1]

[1]Statistics Dept., North Carolina State University, Raleigh, NC, USA

## Abstract

Personalized decision-making, tailored to individual characteristics, is gaining significant attention. The optimal treatment regime aims to provide the best-expected outcome in the entire population, known as the value function. One approach to determine this optimal regime is by maximizing the Augmented Inverse Probability Weighting (AIPW) estimator of the value function. However, the derived treatment regime can be intricate and nonlinear, limiting their use. For clarity and interoperability, we emphasize linear regimes and determine the optimal linear regime by optimizing the AIPW estimator within set constraints.

While the AIPW estimator offers a viable path to estimating the optimal regime, current methodologies predominantly focus on its asymptotic distribution, leaving a gap in studying the linear regime itself. However, there are many benefits to understanding the regime, as pinpointing significant covariates can enhance treatment effects and provide future clinical guidance. In this paper, we explore the asymptotic distribution of the estimated linear regime. Our results show that the parameter associated with the linear regime follows a cube-root convergence to a non-normal limiting distribution characterized by the maximizer of a centered Gaussian process with a quadratic drift. When making inferences for the estimated linear regimes with cube-root convergence in practical scenarios, the standard nonparametric bootstrap is invalid. As a solution, we facilitate the Cattaneo et al. [2020] bootstrap technique to provide a consistent distributional approximation for the estimated linear regimes, validated further through simulations and real-world data applications from the eICU Collaborative Research Database.

## 1 INTRODUCTION

The application of personalized decision-making, which customizes decisions based on individual characteristics, is garnering significant interest across various fields such as economics [Behncke et al., 2009, Turvey, 2017], personalized medicine [Young et al., 2011, Zhang et al., 2020], and reinforcement learning [Jiang and Li, 2016, Munos et al., 2016, Fujimoto et al., 2019]. Personalized medicine, in particular, tailors treatment decisions to the unique attributes of individual patients. A treatment regime takes a patient's specific characteristics as input and determines the appropriate treatment options as output. The optimal treatment regime (also known as an optimal policy, optimal strategy, individualized treatment rule, etc) is the one that maximizes the overall benefit to the patient population, known as the value function. This approach is closely related to a broad body of research in reinforcement learning. Evaluating the expected outcome of patients under a given treatment regime can be viewed as *off-policy evaluation* (OPE) while identifying the optimal treatment regime that yields the highest value function can be considered *off-policy learning* (OFL).

Numerous methods have been developed to identify optimal treatment regimes. One category involves the regression-based approach, which estimates the outcome mean function, referred to as the Q function. It then determines the optimal regime based on this estimated Q function. Techniques in this category include Q-learning [Watkins and Dayan, 1992, Murphy and Littman, 2005, Qian and Murphy, 2011] and A-learning [Murphy, 2003, Shi et al., 2018], which estimate the contrast function of outcome mean functions with varying treatments instead of the original outcome mean function. Another further category is policy search estimation, which is developed by maximizing the value function using either the Inverse Probability Weighting (IPW) estimator, the Augmented Inverse Probability Weighting (AIPW) estimator [Zhang et al., 2012, Zhao et al., 2012], or the Targeted Minimum Loss-Based estimator (TMLE) [van der Laan and Luedtke, 2015, Luedtke and van der

Laan, 2016, Montoya et al., 2021, Poulos et al., 2024]. In this paper, we focus on the policy search estimation for determining the optimal regime, particularly highlighting the AIPW estimator. This estimator is valued for its capability to incorporate machine learning techniques for approximating nuisance parameters. Additionally, it provides double robustness, meaning it remains consistent for the value function if either the outcome mean function or the propensity score model is accurately specified.

Often, the resulting optimal treatment regime can be complex and nonlinear, introducing high variability and posing challenges for practical use. As a result, it's a common practice to seek the optimal regime within a restricted class of regimes. A frequent subset is decision lists [Zhang et al., 2015], where regimes are sequences of decision rules. Each rule is articulated as a series of if-then statements that guide treatment recommendations. These regimes can be explored using tree-based methodologies [Doove et al., 2015, Zhang et al., 2015, Zhao et al., 2015]. In this paper, we focus on linear treatment regimes. These regimes not only pave the way for novel scientific discoveries and hypotheses but are also more accessible for both clinicians and patients. The optimal linear regime is determined by optimizing the AIPW estimator within the pre-specified linear regime constraints.

While some recent papers have made important contributions to estimating the treatment regimes, they focus on studying the asymptotic distributions of the value function [Zhang et al., 2018, Chu et al., 2023a], and have not studied the asymptotic distribution of the estimated treatment regimes. However, there are many benefits to understanding the regime, as identifying significant covariates can enhance treatment effects and offer insights for future clinical practices. To infer the regime directly, it is valuable to determine the asymptotic distributions of the estimated regime. We prove that the estimated parameter indexing the linear treatment regime converges at a cube-root rate to a non-normal limiting distribution that is characterized by the maximizer of a centered Gaussian process with a quadratic drift. Given its cube-root convergence, larger datasets are beneficial for real-world applications. The rise of electronic health records (EHR) — comprehensive digital patient histories maintained over time — offers vast datasets, encompassing demographics, clinical notes, medical histories, and more. This means the requirement for large samples can be readily met using resources like EHR. Hence our study emphasizes large sample scenarios: in simulations, we use a sample size of 20000, and in real data analysis, we utilize the eICU Collaborative Research Database (eICU-CRD) [Goldberger et al., 2000, Pollard et al., 2018, 2019] with 9697 observations.

When constructing confidence intervals for the estimated linear regimes, traditional bootstrap fails for the non-normal cube root convergence estimators [Abrevaya and Huang, 2005, Léger and MacGibbon, 2006, Cattaneo et al., 2020]. Several existing studies have presented consistent

resampling-based distributional approximations for cube-root-type estimators. One category of methods achieves consistency by modifying the distribution used to generate the bootstrap sample. This encompasses techniques like subsampling [Seo and Otsu, 2018], which uses without-replacement subsamples to estimate the cube root estimator distribution; $m$ out of $n$ sampling [Lee and Pun, 2006, Bickel et al., 2012] that employs with-replacement samples for this estimation. Another category modifies the objective function used to construct the bootstrap-based distributional approximation, and then applies the standard nonparametric bootstrap to this altered function, as delineated by Cattaneo et al. [2020]. In our research, we apply the bootstrap method proposed by Cattaneo et al. [2020] to derive the confidence intervals, applying it to both simulation and real data analysis, yielding promising results.

Our main contributions can be summarized as follows:

- Building on the research of Kim and Pollard [1990] and Wang et al. [2018] regarding mean-optimal and quantile-optimal criteria in randomized clinical trials, we establish the linear treatment regime for the AIPW estimator using observational data. We demonstrate that the estimated optimal treatment regime converges at a cube-root rate to a non-normal limiting distribution, characterized by the maximizer of a centered Gaussian process with quadratic drift.

- We utilize the bootstrap technique proposed by Cattaneo et al. [2020] to provide a consistent distributional approximation for the estimated optimal linear regime, addressing the challenges of applying standard nonparametric bootstrap methods to cube-root convergence estimators in practical scenarios.

The rest of the paper is organized as follows. In Section 2, we present the basic setup and introduce the AIPW estimator. Section 3 derives the theorem of the asymptotic distributions of the AIPW estimator and the estimated linear treatment regime. Additionally, this section introduces the algorithm based on the bootstrap method introduced by Cattaneo et al. [2020] for the estimated linear regimes. We conduct simulations in Section 4. Section 5 applies the proposed estimators to an observational study from the eICU-CRD. Finally, we conclude the paper with a discussion in Section 6.

## 2 BACKGROUND

Denote $X \in \mathcal{X} \subset \mathbb{R}^l$ as the vector of pre-treatment covariates, $A \in \{0, 1\}$ as the binary treatment, and $Y \in \mathbb{R}$ as the outcome of interest. Following the potential outcomes framework, let $Y(a)$ be the potential outcome for the subject given the treatment $a, a = 0, 1$. The observed data are then having $n$ independent and identically distributed (i.i.d.) subjects $\{(X_i, A_i, Y_i), i = 1, \ldots, n\}$.

Consider a treatment regime, denoted as $d(X)$, that maps the $l$-dimensional vector $X$ to the set $\{0, 1\}$. As an illustration, if we take $l = 1$ and define the treatment strategy as $d(X) = I(X > 0)$, then for $X = 0.1$, the assigned treatment would be 1. Let $Y(d)$ represent the outcome an individual would achieve when assigned a treatment based on regime $d(X)$ :

$$Y(d) = Y(1)d(X) + Y(0)\{1 - d(X)\}.$$

Further, denote the value function of regime $d(X)$ as

$$V(d) = \mathbb{E}\{Y(d)\} = \mathbb{E}[Y(1)d(X) + Y(0)\{1 - d(X)\}],$$

which is the expected outcome under the regime $d(X)$. Assuming our focus is on a specific collection $D$ of treatment regimes, the optimal treatment regime is the one that maximizes the value function $d^{opt} = \text{argmax}_{d \in D} V(d)$. For practical applications, it's beneficial to consider that is both interpretable and straightforward to implement, hence, in this paper, we focus on the linear regime class $D_\beta = \{d(X; \beta) = I(X^T\beta > 0); \beta \in B\}$, where different treatment regimes are indexed by $\beta$ and $B$ is a compact subset of the parameter space. Notably, $\beta$ is not unique and is equivalent when scaled by scalar multipliers. Therefore, we focus on $B = \{\beta, \|\beta\| = 1\}$, where $\|\cdot\|$ denotes the Euclidean norm. For brevity and clarity, within the context of the linear regime class $D_\beta$, we represent its value function $V(d)$ simply as $V(\beta)$. Denote $\beta_0 = \text{argmax}_{\beta \in B} V(\beta)$, then the optimal regime within regime $D_\beta$ is $d_\beta^{opt} \in D_\beta = d(X; \beta_0) = I(X^T\beta_0 > 0)$.

One of the fundamental challenges to identifying the value function is that $Y(1)$ and $Y(0)$ cannot be observed simultaneously. To overcome this issue, we make the following three common assumptions in the causal inference literature [Rubin, 1978]:

**Assumption 1** $\{Y(0), Y(1)\} \perp\!\!\!\perp A \mid X$ *almost surely, where* $\perp\!\!\!\perp$ *means "independent of".*

**Assumption 2** $Y = Y(1)A + Y(0)(1 - A)$.

**Assumption 3** *There exist constants* $c_1$ *and* $c_2$ *such that* $0 < c_1 \leq Pr(A \mid X) \leq c_2 < 1$ *almost surely.*

Assumption 1 tells us the assignment to treatment is unconfounded. Assumption 2, known as the Stable Unit Treatment Value Assumption (SUTVA), suggests a lack of interference. This means that the potential outcomes for one individual remain unaffected by the treatments received or the potential outcomes of other individuals. Under Assumption 1-2, the conditional mean of the potential outcome $Y(a)$ can be represented in terms of observed data. Specifically, the conditional outcome mean function $\mu_A(X) = \mathbb{E}\{Y(A) \mid X\} = \mathbb{E}(Y \mid X, A)$. Assumption 3 implies a sufficient overlap of the covariate distribution between the treatment groups.

Denote the propensity score as $e(X) = Pr(A \mid X)$. Given the regime $d(X; \beta)$ and under Assumptions 1-3, it is imperative to highlight that

$$\mathbb{E}\left[\frac{I\{A = d(X; \beta)\}}{\rho(A \mid X)}\{Y - \mu_d(X; \beta)\}\right] = 0,$$

where

$$\rho(A \mid X) = e(X)A + \{1 - e(X)\}(1 - A),$$
$$\mu_d(X; \beta) = \mu_1(X)I\{d(X; \beta) = 1\}$$
$$+ \mu_0(X)I\{d(X; \beta) = 0\}.$$

Consequently, the value function can be expressed as

$$V(\beta)$$
$$= \mathbb{E}[Y\{d(X; \beta)\}]$$
$$= \mathbb{E}\{\mu_d(X; \beta)\} + \mathbb{E}\left[\frac{I\{A = d(X; \beta)\}}{\rho(A \mid X)}\{Y - \mu_d(X; \beta)\}\right].$$

Further, if we denote

$$v(X, A, Y; \beta)$$
$$= \frac{I\{A = d(X; \beta)\}}{\rho(A \mid X)}\{Y - \mu_d(X; \beta)\} + \mu_d(X; \beta),$$

then $V(\beta)$ can be written as $V(\beta) = \mathbb{E}\{v(X, A, Y; \beta)\}$.

Building upon this foundational understanding, Zhang et al. [2012] proposed an AIPW estimator $\hat{V}_n(\beta)$ for the value function $V(\beta)$ as

$$\hat{V}_n(\beta) = \frac{1}{n}\sum_{i=1}^{n}\hat{v}(X_i, A_i, Y_i; \beta),$$

where

$$\hat{v}(X_i, A_i, Y_i; \beta)$$
$$= \frac{I\{A_i = d(X_i; \beta)\}}{\hat{\rho}(A_i \mid X_i)}\{Y_i - \hat{\mu}_d(X_i; \beta)\} + \hat{\mu}_d(X_i; \beta),$$

and $\hat{e}(X), \hat{\mu}_A(X)$ are the estimates for $e(X)$ and $\mu_A(X)$. $\hat{\rho}(A \mid X)$ and $\hat{\mu}_d(X; \beta)$ are derived by substituting the estimates $\hat{e}(X)$ and $\hat{\mu}_A(X)$ into $\rho(A \mid X)$ and $\mu_d(X; \beta)$, respectively. Further, denote $\hat{\beta} = \text{argmax}_{\beta \in B} \hat{V}_n(\beta)$.

When using parametric models to estimate $\mu_A(X)$ (for $A = 0, 1$) or $e(X)$, the AIPW estimator consistently approximates $V(\beta)$ if either the posited parametric model for $\mu_A(X)$ (for $A = 0, 1$) or for $e(X)$ is correctly specified. Despite this advantageous property, real-world situations frequently pose difficulties in correctly implementing parametric models. Recently, machine learning methods have gained traction. Various semi-parametric or non-parametric machine learning algorithms can be utilized to consistently estimate the unknown functions $e(X)$ and $\mu_A(X)$ for $A = 0, 1$. Consequently, in this paper we focus on the use of semi-parametric or non-parametric machine learning models for

estimating $e(X)$ and $\mu_A(X)$ for $A = 0, 1$. Notably, these findings can be extended to the scenario where $e(X)$ and $\mu_A(X)$ are estimated via parametric models, and a detailed discussion is in Section 6.

# 3 METHODS

In this section, we delve into several key topics. Subsection 3.1.1 focuses on the derivation of the asymptotic distributions for the AIPW estimator. Meanwhile, subsection 3.1.2 explores the asymptotic distributions of the estimated linear treatment regimes. Additionally, subsection 3.2 introduces the algorithm based on the bootstrap method proposed by Cattaneo et al. [2020] for the estimated linear regime.

## 3.1 THEOREM

### 3.1.1 Asymptotic distribution of the AIPW estimator

We begin by outlining the theorem related to the asymptotic properties of $\hat{V}_n(\hat{\beta})$. First, we assume the following regularity conditions:

**Assumption 4** *The support of $X$ and $Y$ are bounded.*

**Assumption 5** *The function $\mu_a(x)$ is smooth, and continuously differentiable and bounded for all $(x, a)$.*

**Assumption 6** *The optimal treatment regime $\beta_0 \in B$ satisfying $\|\beta_0\| = 1$, is unique.*

**Assumption 7**

$$\left[ \mathbb{E} \left\{ \hat{e}(X) - e(X) \right\}^2 \right]^{1/2} \sum_{a=0}^{1} \left[ \mathbb{E} \left\{ \hat{\mu}_a(X) - \mu_a(X) \right\}^2 \right]^{1/2}$$
$$= o_p(n^{-1/2}).$$

Assumptions 4-5 are standard regularity conditions used to establish the convergence results. Assumption 6 is an identifiability condition for $\beta_0$ and ensures the true targeted optimal regime $d(X; \beta_0)$ is uniquely defined, similar to Wang et al. [2018]. To meet the criteria of Assumption 7, one approach entails ensuring both $\sum_{a=0}^{1} \left[ \mathbb{E} \left\{ \hat{\mu}_a(X) - \mu_a(X) \right\}^2 \right]^{1/2} = o_p(n^{-1/4})$ and $\left[ \mathbb{E} \left\{ \hat{e}(X) - e(X) \right\}^2 \right]^{1/2} = o_p(n^{-1/4})$. In this context, purely nonparametric estimators, such as kernel or nearest-neighbor methods, are generally not viable due to their slower convergence rate, specifically below $o_p(n^{-1/4})$. However, certain semi-parametric models, like generalized additive models, can attain rates of $n^{-2/5}$. For a comprehensive list of estimators that can reach $o_p(n^{-1/4})$ convergence

rates, we refer to the book [Horowitz, 2009] and the review article [Kennedy, 2016].

The result is as follows:

**Theorem 1** *Under Assumptions 1-7, as $n \to \infty$, we have*

1. *$\|\hat{\beta} - \beta_0\| = O_p(n^{-1/3})$.*
2. *$\sqrt{n} \left\{ \hat{V}_n(\hat{\beta}) - V(\beta_0) \right\} \xrightarrow{D} \mathcal{N}(0, \sigma^2),$*

*where $\xrightarrow{D}$ represents convergence in distribution, and*

$$\sigma^2 = \mathbb{E} \left[ \frac{I\{A_i = d(X_i; \beta_0)\}}{\rho(A_i \mid X_i)} \{Y_i - \mu_d(X_i; \beta_0)\} + \mu_d(X_i; \beta_0) - V(\beta_0) \right]^2.$$

The asymptotic distribution results provide valuable insights for making inferences regarding $V(\beta_0)$. It's important to highlight that while $\hat{V}_n(\hat{\beta})$ converges at a $\sqrt{n}$-consistent rate, the convergence rate of regime $\hat{\beta}$ is $n^{1/3}$. This deviates from many established statistical theorems, such as the central limit theorem, which operates on the square root rate $O_p(n^{-1/2})$. Despite the extensive exploration of the $\sqrt{n}$-consistent AIPW estimator $\hat{V}_n(\hat{\beta})$ and its various modified versions, the asymptotic distribution of $\hat{\beta}$ remains relatively unexplored due to its $n^{1/3}$ convergence rate. In the following section, we explore the asymptotic distribution of $\hat{\beta}$. Wang et al. [2018] investigated the linear treatment regime for both the mean-optimal and quantile-optimal criteria in randomized clinical trials. We build upon their findings to deduce the linear treatment regime for the AIPW estimator using observational data.

### 3.1.2 Asymptotic distribution of the estimated linear regime

Kim and Pollard [1990] deduced the asymptotic distribution related to cube root convergence, however, the result of Kim and Pollard [1990] is not directly transferrable because $\hat{V}_n(\hat{\beta})$ incorporates the estimated $\hat{e}(X)$ and $\hat{\mu}_A(X)$ for $A = 0, 1$. Hence, we examine the conditions outlined in Kim and Pollard [1990] sequentially in the proof of Theorem 2, as detailed in the Supplementary material. First, we introduce the following conditions:

**Assumption 8** *$X$ has a continuously differentiable density $f(\cdot)$ and that the angular components of $X$, considered as a random element of the unit sphere $S$ in $R^l$, has a bounded continuous density with respect to surface measure on $S$.*

**Assumption 9**

$$H = \int \left\{x^{\mathrm{T}}\beta_0 = 0\right\} \left\{\dot{f}(x)h(x) + f(x)\dot{h}(x)\right\}^{\mathrm{T}} \beta_0 x x^{\mathrm{T}} d\sigma,$$

*and $H > 0$, where $\sigma$ is the surface measure on the hyperplane $\{X : X^{\mathrm{T}}\beta_0 = 0\}$ and $h(x) = \mathbb{E}\{Y(1) - Y(0) \mid X = x\}$. $\dot{f}(x)$ and $\dot{h}(x)$ denote the first-gradient with respect to $x$.*

**Assumption 10** $\sup_{x \in \mathcal{X}} |\hat{e}(X) - e(X)| = o_p(n^{-1/3})$, *and* $\sup_{x \in \mathcal{X}} |\hat{\mu}_A(X) - \mu_A(X)| = o_p(n^{-1/3})$ *for $A = 0, 1$.*

Assumptions 8-9 are technical conditions for evaluating the first and second-order derivatives of the value function and the kernel covariance which are used to characterize the asymptotic distribution of $\hat{\beta}$, similar in Example 6.4 in Kim and Pollard [1990] and Wang et al. [2018]. Assumption 8 aids in deriving the first derivative of $V(\beta_0)$. Under Assumption 9, as deduced from the proof of Theorem 2, the matrix $-H$ represents the second order derivative of value function $V(\beta_0)$ at $\beta = \beta_0$,

$$H = \int \left\{x^{\mathrm{T}}\beta_0 = 0\right\} \left\{\dot{f}(x)h(x) + f(x)\dot{h}(x)\right\}^{\mathrm{T}} \beta_0 x x^{\mathrm{T}} d\sigma$$
$$= -\partial^2 V(\beta_0)/\partial\beta\partial\beta^{\mathrm{T}}.$$

Ensuring $H$ is positively definite is one crucial condition in Kim and Pollard [1990] for the asymptotic distribution of $n^{1/3}(\hat{\beta} - \beta_0)$.

Assumption 10 requires $\hat{e}(X)$ and $\hat{\mu}_A(X)$ to uniformly converge to $e(X)$ and $\mu_A(X)$ at a rate of $o_p(n^{-1/3})$. It's noticeable that the requisite convergence rate of $\hat{e}(X)$ and $\hat{\mu}_A(X)$ for $A = 0, 1$ under Assumption 10 is faster than the approach suggested following Assumption 7. Technically speaking, the reason for the faster $o_p(n^{-1/3})$ uniform convergence rate in Assumption 10 is to meet the first condition set out in Kim and Pollard [1990], ensuring the existence of the cube root asymptotic distribution of $\hat{\beta}$ exists. A perspective to understand this is by recognizing that when developing the $\sqrt{n}$-consistent rate asymptotic distribution of the value function $\hat{V}_n(\hat{\beta})$, it is only requiring the $\|\hat{\beta} - \beta_0\| = O_p(n^{-1/3})$, however, given $\|\hat{\beta} - \beta_0\| = O_p(n^{-1/3})$, it still needs further restriction to guarantee the asymptotic distribution of $\hat{\beta}$. That's why the conditions are more strict in Assumption 10 than the approach suggested following Assumption 7.

To guarantee that there are estimators reaching this $o_p(n^{-1/3})$ uniform convergence rate for estimating $\hat{e}(X)$ and $\hat{\mu}_A(X)$ for $A = 0, 1$, first assuming that both $e(X)$ and $\mu_A(X)$ for $A = 0, 1$ belong to the function class $\Sigma_s$, where $\Sigma_s$ represents the Holder classes of smoothness order $s$. For $s \in (0, 1]$, the Holder class $\Sigma_s$ is defined as the set of all

functions $f: \mathcal{X} \to \mathbb{R}$ such that for $C > 0$,

$$|f(x) - f(\tilde{x})| \leq C \left\{\sum_{j=1}^{l}(x_j - \tilde{x}_j)^2\right\}^{s/2}$$

for all $x, \tilde{x} \in \mathcal{X}$. For $s > 1$, $\Sigma_s$ is defined as follows. For any $\alpha = (\alpha_1, \cdots, \alpha_l)$ of nonnegative integers, denote $D^{\alpha} = \partial_{x_1}^{\alpha_1} \cdots \partial_{x_l}^{\alpha_l}$. Then $\Sigma_s$ is the set of all functions $f : \mathcal{X} \to \mathbb{R}$ such that $f$ is $[s]$ times continuously differentiable and for some $C > 0$,

$$|D^{\alpha}f(x) - D^{\alpha}f(\tilde{x})| \leq C \left\{\sum_{j=1}^{l}(x_j - \tilde{x}_j)^2\right\}^{(s-[s])/2}$$

and $|D^{\beta}f(x)| \leq C$ hold for all $x, \tilde{x} \in \mathcal{X}$, where $\alpha = (\alpha_1, \ldots, \alpha_l)$ and $\beta = (\beta_1, \ldots, \beta_l)$ are nonnegative integers satisfying $\alpha_1 + \cdots \alpha_l = [s]$ and $\beta_1 + \cdots \beta_l \leq [s]$. Given these assumptions, the optimal uniform rate of convergence for $\hat{e}(X)$ and $\hat{\mu}_A(X)$ (for $A = 0, 1$) is $O_p\left\{(\ln n/n)^{s/(2s+l)}\right\}$ [Stone, 1982]. This rate can be achieved using various estimators, such as series estimators [Belloni et al., 2015] and local polynomial (kernel) estimators [Takezawa, 2005]. Additionally, when $e(X)$ and $\mu_A(X)$ (for $A = 0, 1$) are determined using estimators that meet this optimal uniform rate and if $e(X)$ and $\mu_A(X)$ adhere to the condition $s > l$, the uniform rate achieves $o_p(n^{-1/3})$. In such scenarios, it's feasible to employ estimators that realize this $o_p(n^{-1/3})$ rate to estimate $e(X)$ and $\mu_A(X)$, where $A = 0, 1$.

The theorem is as follows:

**Theorem 2** *Under Assumptions 1-6, and Assumptions 8-10, we have*

$$n^{1/3}(\hat{\beta} - \beta_0) \xrightarrow{D} \operatorname{argmax}_t \left\{-\frac{1}{2}t^{\mathrm{T}}Ht + W(t)\right\},$$

*where $\xrightarrow{D}$ represents converge in distribution. $H = -\partial^2 V(\beta_0)/\partial\beta\partial\beta^{\mathrm{T}}$ is a $l \times l$ positively definite matrix and $W(t)$ is a zero-mean Gaussian process with continuous sample paths and covariance kernel $C(\cdot, \cdot)$. The expressions for $C(\cdot, \cdot)$ is in the Supplementary material.*

## 3.2 BOOTSTRAP ALGORITHM

The distribution result in Theorem 2 sheds light on constructing inference on $\beta_0$ as we develop in subsection 3.1.2. While we theoretically determine the asymptotic distribution of $\hat{\beta}$, applying these theoretical results for inference in practice can be challenging. A straightforward approach in real-world scenarios might involve using bootstrap methods to sample the distribution of $\hat{\beta}$. However, the standard nonparametric bootstrap is often inadequate in approximating the cube root distribution. We provide a straightforward demonstration below. Let's define

$V_n(\beta) = 1/n \sum_{i=1}^{n} v(X_i, A_i, Y_i; \beta)$. According to Kim and Pollard [1990], the cube root convergence for $\hat{\beta}$ can be written as $n^{1/3}(\hat{\beta} - \beta_0) = \operatorname{argmax}_{t \in \mathbb{R}^l} \{\hat{W}(t) + \mathcal{V}(t)\}$,

$$\hat{W}(t) = n^{2/3} \left\{ V_n(\beta_0 + tn^{-1/3}) - V_n(\beta_0) \right.$$
$$\left. - V(\beta_0 + tn^{-1/3}) + V(\beta_0) \right\}$$

is a zero-mean random process and asymptotically converges to $W(t)$ and

$$\mathcal{V}(t) = n^{2/3} \left\{ V(\beta_0 + tn^{-1/3}) - V(\beta_0) \right\}$$

asymptotically converges to $-t^T H t / 2$, where $H = -\partial^2 V(\beta_0)/\partial\beta\partial\beta^T$. The standard nonparametric bootstrap can replicate the shape of $\hat{W}(t)$, however, it fails to replicate the shape of $\mathcal{V}(t)$, which results in the inconsistency of the standard nonparametric bootstrap [Abrevaya and Huang, 2005, Léger and MacGibbon, 2006, Cattaneo et al., 2020].

To address this, a refined bootstrap methodology was proposed by Cattaneo et al. [2020] to more accurately approximate the distribution for $\hat{\beta}$. This method alters the objective function to ensure that the bootstrap version of each empirical process counterpart has a mean resembling its large sample version. Specifically, Cattaneo et al. [2020] reshaped the original objective function $\hat{v}(X_i, A_i, Y_i; \beta)$ to

$$\tilde{v}(X_i, A_i, Y_i; \beta)$$
$$= \hat{v}(X_i, A_i, Y_i; \beta) - \hat{V}_n(\beta) - \frac{1}{2}(\hat{\beta} - \beta)^T H_n(\hat{\beta} - \beta),$$

where $H_n$ serves as an approximation of $H$. This adjustment ensures the convergence of the bootstrap versions to their population counterparts in large samples. For $\hat{\beta}$, the bootstrap samples are represented as $\hat{\beta}^*$, given by

$$\hat{\beta}^* = \operatorname{argmax}_{\beta \in B} \hat{V}_n^*(\beta),$$
$$\hat{V}_n^*(\beta) = \frac{1}{n} \sum_{i=1}^{n} \tilde{v}(X_i^*, A_i^*, Y_i^*; \beta),$$

where $\{(X_i^*, A_i^*, Y_i^*), i = 1, \ldots, n\}$ are random samples from the empirical distribution $(X, Y, A)$. It's important to highlight that during this bootstrap procedure, the nuisance parameters can either be refitted using the samples $\{(X_i^*, A_i^*, Y_i^*), i = 1, \ldots, n\}$ during the bootstrap process or retain their initially estimated values from the original datasets. Both approaches meet the conditions outlined in Cattaneo et al. [2020]. Cattaneo et al. [2020] proved that under certain regularity conditions, $n^{1/3}(\hat{\beta}^* - \hat{\beta}) \to \operatorname{argmax}_t \{-\frac{1}{2} t^T H t + W(t)\}$ in distribution, therefore guarantees consistency of the bootstrap samples. Algorithm 1 outlines the detailed procedure for bootstrapping samples. Regarding computational requirements, the complexity of

---

**Algorithm 1** The proposed bootstrap algorithm.

Step 1: Using the sample $(X_i, Y_i, A_i)$, compute $\hat{\beta}$ by approximately maximizing $\hat{V}_n(\beta)$.

Step 2: Using $\hat{\beta}$ and $(X_i, Y_i, A_i)$, compute $H_n$, where each $(k, l)$ element in $H_n$ is defined as:

$$H_{n,kl} = -\frac{1}{4\epsilon_n^2} \left\{ \hat{V}_n(\hat{\beta} + e_k \epsilon_n + e_l \epsilon_n) \right.$$
$$- \hat{V}_n(\hat{\beta} + e_k \epsilon_n - e_l \epsilon_n) - \hat{V}_n(\hat{\beta} - e_k \epsilon_n + e_l \epsilon_n)$$
$$\left. + \hat{V}_n(\hat{\beta} - e_k \epsilon_n - e_l \epsilon_n) \right\},$$

where $e_k$ is the $k$-th unit vector in $\mathbb{R}^l$ and $\epsilon_n$ is a positive tuning parameter. $H_n$ is a consistent estimator of $H$.

Step 3: Using $\hat{\beta}$, $H_n$, and the bootstrap sample $\{(X_i^*, A_i^*, Y_i^*), i = 1, \ldots, n\}$, compute $\hat{\beta}^*$ by approximating maximizing $\hat{V}_n^*(\beta)$.

Step 4: Repeat Step 3 to generate draws from the distribution $n^{1/3}(\hat{\beta}^* - \hat{\beta})$.

---

this process is expressed as $O(K\tilde{B})$, where $\tilde{B}$ is the size of bootstrap samples, and $K$ denotes the algorithm's complexity for obtaining the estimate $\hat{\beta}$ given $\hat{V}_n(\beta)$. Specifically, employing a genetic algorithm [Katoch et al., 2021] introduces a complexity of $K = O(GNn)$, with $G$ indicating the number of iterations and $N$ the population size. Incorporating this methodology, we utilized it to determine the 95% confidence interval for $\hat{\beta}$ in our simulations.

A key element of their approach revolves around the tuning parameter, $\epsilon_n$. Although Cattaneo et al. [2020] suggested an optimal value for $\epsilon_n$ that minimizes the approximate Mean Squared Error, this ideal value incorporates both the $H$ matrix and the covariance kernel $C(\cdot, \cdot)$. Determining the best $\epsilon_n$ involves estimating both $H$ and $C(\cdot, \cdot)$, a process that is intricate and not feasible in our context.

Beyond this method, there exist other strategies that guarantee consistency for cube root convergence estimators by adjusting the distribution from which the bootstrap sample is drawn. Examples include subsampling methods, [Seo and Otsu, 2018], which use subsamples without replacement, and the $m$ out of $n$ sampling techniques [Lee and Pun, 2006, Bickel et al., 2012], which rely on samples with replacement for their estimates. Hong and Li [2020] introduced a numerical bootstrap technique where bootstrap samples are determined by the maximizer of the linear combination of the empirical distribution and the bootstrapped empirical process.

However, both the $m$ out of $n$ approach and subsampling necessitate that the count of bootstrap samples be $o_p(n)$. In practical applications, when maximizing a non-regular objective function to obtain the cube root convergence estimator, a limited count in the bootstrap samples can lead

to results that deviate from true values, compromising the outcomes. The numerical bootstrap method [Hong and Li, 2020], on the other hand, is also contingent on a tuning parameter.

# 4 SIMULATION

In this section, we conduct one simulation study with $T = 100$ simulation times. To effectively achieve the cube root convergence in finite samples, a substantial number of observations is essential. Thus, in each simulation time, we generate $n = 20000$ observations $(X_i, Y_i, A_i), i = 1, \ldots, n$, where $X_i = (1, X_{i1}, X_{i2})^{\mathrm{T}}$ and $X_{i1}$ and $X_{i2}$ are independent following the uniform distribution on $[1 - \sqrt{3}, 1 + \sqrt{3}]$; given $X_i$, the binary treatment indicator $A_i$ satisfies $\mathrm{logit}\{e(X_i)\} = -1.0 + 0.8X_{i1} + 0.8X_{i2}$, where $\mathrm{logit}(u) = \log\{u/(1-u)\}$; and outcomes are generated as $Y_i = 2 - 1.5X_{i1} - 1.5X_{i2} + A \times (2X_{i1} + X_{i2}) + \epsilon_i, \epsilon_i \sim \mathcal{N}(0, 1)$. To estimate $\mathrm{logit}\{e(X)\}$ and $\mu_A(X)$ for $A = 0, 1$, we apply the generalized additive model (GAM) using the cubic smoothing spline for each univariate covariate. From Eggermont et al. [2001], the cubic smoothing spline can achieve the optimal uniform rate, therefore $\hat{e}(X)$ and $\hat{\mu}_A(X)$ (for $A = 0, 1$) can uniformly converge to $e(X)$ and $\mu_A(X)$ for $A = 0, 1$ with the rate $o_p(n^{-1/3})$.

For each individual, the optimal treatment regime is given by $I\{\mu_1(X_i) > \mu_0(X_i)\}$, therefore in our example the optimal regime is $d(X_i) = I(2X_{i1} + X_{i2} > 0)$, which is a linear regime. The optimization process is facilitated using the `genoud` function in the R package `rgenoud` [Mebane Jr and Sekhon, 2011]. To achieve the uniqueness, we impose the restriction $\|\beta\| = 1$, and the true optimal linear rule $\beta_0 = (\beta_{01}, \beta_{02})^{\mathrm{T}} = (0.894, 0.447)^{\mathrm{T}}$.

To determine the 95% confidence interval for $\hat{\beta}$, we incorporate the Cattaneo et al. [2020] Bootstrap method, with the use of 400 bootstrap samples. In each bootstrap sample, we refit both $e(X)$ and $\mu_A(X)$ to obtain the AIPW estimator and derive the estimated linear regimes. Identifying the optimal tuning parameter $\epsilon_n$, which minimizes the approximate Mean Squared Error, requires the estimated $H$ matrix and the covariance kernel $C(\cdot, \cdot)$. Considering the intricate nature and challenges presented in this context, we evaluate multiple $\epsilon_n$ values to identify the most suitable one.

Table 1 presents the estimates for $\beta_{01}$ and $\beta_{02}$ (denoted as "Est") and their 95% quantile confidence interval length (denoted as "Length") and coverage rate (denoted as "Coverage"). The findings suggest that an optimal $\epsilon_n$ is approximately 0.5, as it achieves a 95% coverage rate and the shortest confidence interval length. This indicates that, with a judicious selection of $\epsilon_n$, the Cattaneo et al. [2020] bootstrap method can present a reasonable inference of the $\hat{\beta}$.

Table 1: Simulation results under different tuning parameters $\epsilon_n$ based on 100 Monte Carlo times with 400 bootstrap samples in each simulation time.

| Est | | $\epsilon_n$ | | | | | |
| --- | --- | 0.05 | 0.1 | 0.2 | 0.5 | 0.7 | 0.9 |
| $\beta_{01}$ 0.895 | Coverage | 0.750 | 0.900 | 1 | 0.95 | 1 | 1 |
| | Length | 0.139 | 0.894 | 0.984 | 0.086 | 0.135 | 0.235 |
| $\beta_{02}$ 0.443 | Coverage | 0.740 | 0.920 | 1 | 0.95 | 1 | 1 |
| | Length | 0.207 | 0.864 | 1.170 | 0.176 | 0.261 | 0.416 |

# 5 REAL DATA APPLICATION

We demonstrate our proposed approach using data sourced from the eICU-CRD, a multi-center repository of anonymized health records spanning across the United States from 2014 to 2015 [Goldberger et al., 2000, Pollard et al., 2018, 2019].

We consider the 9 baseline covariates: age (years), Body Mass Index (BMI), derived by dividing admission weight (kg) by the square of admission height (meters), admission temperature (Temp) value (Celsius), glucose level (mg/dL), blood urea nitrogen (BUN) amount (mg/dL), creatinine amount (mg/dL), white blood cell (WBC) count (K/uL), bilirubin (mg/dL), mean blood pressure (BP) level (mmHg). A treatment value of 1 indicates the patient was administered vasopressor, while a value of 0 suggests other medical interventions. We consider the cumulative balance (mL) as the outcome of interest. A positive cumulative balance means the fluid intake exceeds the output, leading to a condition called hypervolemia or fluid overload. Excess fluid can strain the heart, potentially causing heart failure [Gologorsky and Roy, 2020], rapid decline in kidney function, and an increased need for kidney replacement therapy [Palmer and Clegg, 2020]. Conversely, a negative balance implies the patient's output exceeded their intake, labeled as hypovolemia or fluid deficit. Severe hypovolemic shock can result in mesenteric and coronary ischemia that can cause abdominal or chest pain [Taghavi et al., 2022]. For our study, we use $Y = -|\text{cumulative balance}|$ (CB) as the outcome, where a higher value is preferable.

After filtering the abnormal values, a total of 9697 observations remained. Table 2 summarizes the mean and the standard deviation of the outcome and covariates in the samples. To utilize the bootstrap method proposed by Cattaneo et al. [2020], we generate 100 bootstrap samples to derive the 95% confidence intervals for the estimated linear regimes and conduct sensitivity analysis across various $\epsilon_n$ values, specifically within the set $\{0.3, 0.5, 0.7\}$. Due to the widest confidence interval ranges observed at $\epsilon_n = 0.7$, which results in no findings for significant covariates, we focus on presenting results for $\epsilon_n = 0.3$ and 0.5 in the main text. The corresponding results for $\epsilon_n = 0.7$ are included in the supplementary materials. Detailed estimates and confi-

Table 2: Mean and the standard deviation (denoted as "sd") of the outcome and covariates in the samples.

|  | age | BMI | Temp | Glucose | BUN |
|---|---|---|---|---|---|
| mean | 64.95 | 28.94 | 36.04 | 149.73 | 29.55 |
| sd | 15.32 | 7.63 | 4.71 | 107.25 | 26.72 |
|  | creatinine | WBC | bilirubin | BP | $-|CB|$ |
| mean | 1.52 | 12.23 | 0.33 | 75.83 | -5929.21 |
| sd | 2.01 | 11.70 | 2.42 | 42.11 | 5822.28 |

Table 3: Estimates for the linear regime (denoted as "est"), the corresponding $95\%$ confidence intervals (denoted as "CI") and the confidence interval lengths (denoted as "Length") when $\epsilon_n = 0.3$ and $0.5$.

|  | Est |  | $\epsilon_n$ | |
|---|---|---|---|---|
|  |  |  | 0.3 | 0.5 |
| Int | 0.489 | CI | (-0.609, 0.765) | (-0.408, 0.675) |
|  |  | Length | 1.374 | 1.082 |
| age | 0.254 | CI | (-0.129, 0.313) | (-0.374, 0.445) |
|  |  | Length | 0.442 | 0.819 |
| BMI | 0.087 | CI | (-0.162, 0.245) | (-0.123, 0.501) |
|  |  | Length | 0.407 | 0.624 |
| **Temp** | 0.424 | CI | **(0.004, 0.637)** | (-0.009, 0.681) |
|  |  | Length | 0.633 | 0.690 |
| Glucose | -0.382 | CI | (-0.422, 0.020) | (-0.497, 0.066) |
|  |  | Length | 0.442 | 0.564 |
| BUN | -0.279 | CI | (-0.416, 0.019) | (-0.469, 0.235) |
|  |  | Length | 0.435 | 0.704 |
| creatinine | -0.162 | CI | (-0.422, 0.273) | (-0.286, 0.241) |
|  |  | Length | 0.694 | 0.527 |
| **WBC** | 0.486 | CI | (-0.149, 0.721) | **(0.038, 0.971)** |
|  |  | Length | 0.869 | 0.934 |
| bilirubin | 0.133 | CI | (-0.586, 0.587) | (-0.453, 0.697) |
|  |  | Length | 1.173 | 1.150 |
| BP | 0.072 | CI | (-0.177, 0.224) | (-0.317, 0.268) |
|  |  | Length | 0.402 | 0.585 |

dence intervals for $\epsilon_n = 0.3$ and $0.5$ are shown in Table 3, where the intercept is denoted as "Int"." Both $\epsilon_n = 0.3$ and $\epsilon_n = 0.5$ produce comparable confidence interval lengths. Given the challenges in pinpointing the optimal $\epsilon_n$ value in real-world scenarios, it's prudent to consider the results from both $\epsilon_n = 0.3$ and $\epsilon_n = 0.5$, especially since their confidence interval lengths are similar. For $\epsilon_n = 0.3$, temperature stands out as a significant covariate, exerting a positive impact on the linear regime. This aligns with clinical understanding, as sepsis often leads to fever [Schortgen, 2012]. On the other hand, with $\epsilon_n = 0.5$, the white blood cell (WBC) count becomes a significant covariate, also positively affecting the linear regime. This is consistent with medical knowledge, as sepsis usually produces an elevated white blood cell count [Munford, 2006].

## 6 DISCUSSION

In this paper, we focus on the linear regimes. We present the asymptotic properties of the AIPW estimators and explore the non-normal asymptotic distribution of the estimated linear regime with the cube root convergence rate. Recognizing that the standard nonparametric bootstrap fails to approximate the cube root distribution, we implement the bootstrap method in Cattaneo et al. [2020] to provide a valid bootstrap sample for the linear regime converging to the non-normal cube root distribution.

It's important to highlight that while our primary focus is on the semi-parametric models and non-parametric models for estimating $\mu_A(X)$ (for $A = 0, 1$) and $e(X)$, the findings can be easily extended to the parametric models of $\mu_A(X)$ and $e(X)$. Parametric models can achieve a convergence rate of $O_p(n^{-1/2})$, which is faster than the rates observed in both semi-parametric and non-parametric models. Theorem 1 remains valid provided at least one model among $\mu_A(X)$ and $e(X)$ is correctly specified [Chu et al., 2023a]. However, to derive the cube root distribution of $\hat{\beta}$, two prerequisites are essential. Firstly, the nuisance parameters in $\mu_A(X)$ and $e(X)$ must achieve a convergence rate of $o_p(n^{-1/3})$, a feat readily accomplished by parametric models with a convergence rate of $O_p(n^{-1/2})$. Secondly, both the $e(X)$ and $\mu_A(X)$ models must be correctly specified, a stipulation that might pose challenges in real-world scenarios. Conversely, when centering on the parametric models, the bootstrap approach in Cattaneo et al. [2020], which does not consider the nuisance parameters, remains inapplicable. As an alternative, one might look into the $m$ out of $n$ sampling method [Lee and Pun, 2006], which remains valid even in the presence of nuisance parameters.

Selecting the optimal $\epsilon_n$ can be challenging for the bootstrap method in Cattaneo et al. [2020]. For practical applications, we recommend choosing $\epsilon_n$ values that correspond to local minima in the lengths of confidence intervals. This strategy guided our choice of $\epsilon_n = 0.5$ for the simulations in Section 4, where this value yielded the shortest confidence intervals with a coverage rate close to 95%. Similarly, for the real data analysis in Section 5, after evaluating $\epsilon_n$ values of $\{0.3, 0.5, 0.7\}$, we identified $\{0.3, 0.5\}$ as the local minima for most variables tested.

There are some extensions we will consider in future work. First, our study centers on a single-stage treatment regime with two treatment options. While suitable for some research scenarios, it doesn't encompass the broader complexities of treatment pathways. Extending our analysis to multi-stage treatment regimes is possible by utilizing proof techniques akin to those employed by Wang et al. [2018]. This approach offers a promising direction for further research.

Second, our attention is centered on linear regimes, valued for their interpretability and ease of communication.

While decision lists [Zhang et al., 2015] are another popular regime, current literature mainly investigates value function estimates and offers efficient computational algorithms for estimating these decision lists [Doove et al., 2015, Zhang et al., 2015, Zhao et al., 2015]. However, there's a noticeable gap in the literature regarding the inference of these regimes. Exploring the inference for decision lists could be a compelling extension.

Third, our study predominantly focuses on the univariate continuous outcome $Y$. Yet, the breadth of our investigation can be extended to include diverse outcomes such as survival rates, binary results, and counting processes. Furthermore, our proposed inference framework applies readily to the transfer learning approach of optimal linear regimes from a source population to a target population [Chu et al., 2023a,b, Colnet et al., 2024, Lee et al., 2022, 2023, 2024a,b, Wu and Yang, 2022, 2023].

Finally, the foundational assumptions in our study include the absence of unmeasured confounders (Assumption 1), SUTVA (Assumption 2), and positivity (Assumption 3). The violation of either assumption will lead to biases in our results. Given this, it is essential for subsequent studies to conduct sensitivity analyses to scrutinize the assumptions against unmeasured confounders and SUTVA. Regarding the positivity assumption, Zhao et al. [2024] introduced a positivity-free policy learning, which can be our future extension.

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

# Inference for Optimal Linear Treatment Regimes in Personalized Decision-making
## (Supplementary Material)

**Yuwen Cheng**[1]                    **Shu Yang**[1]

[1]Statistics Dept., North Carolina State University, Raleigh, NC, USA

The supplementary material is structured as follows: Section A and Section B provide proofs for the main theorems. Section C displays additional tables for the confidence intervals for the estimated linear regimes in the eICU-CRD datasets when $\epsilon_n = 0.7$.

## A    PROOF OF THEOREM 1

We derive the asymptotic distribution of $\hat{V}_n(\hat{\beta})$. We follow the similar proof in Chu et al. [2023a]

First, we show

$$\hat{V}_n(\beta) = V_n(\beta) + o_p(n^{-1/2}), \ \forall \beta.$$

We define a middle term as

$$\bar{v}_n(\beta) = \frac{1}{n} \sum_{i=1}^{n} \left[ \frac{I\{A = d(X_i; \beta)\}}{\rho(A_i \mid X_i)} \{Y - \hat{\mu}_d(X_i; \beta)\} + \hat{\mu}_d(X_i; \beta) \right],$$

and show $\hat{V}_n(\beta) = \bar{v}_n(\beta) + o_p(n^{-1/2})$ and $\bar{v}_n(\beta) = V_n(\beta) + o_p(n^{-1/2})$.

$$
\begin{aligned}
&\hat{V}_n(\beta) - \bar{v}_n(\beta) \\
=& \frac{1}{n} \sum_{i=1}^{n} \left( \left[ \frac{I\{A = d(X_i; \beta)\}}{\hat{\rho}(A_i \mid X_i)} - \frac{I\{A = d(X_i; \beta)\}}{\rho(A_i \mid X_i)} \right] \{Y - \hat{\mu}_d(X_i; \beta)\} \right) \\
=& \frac{1}{n} \sum_{i=1}^{n} \left( (2A_i - 1)\{e(X_i) - \hat{e}(X_i)\} \left[ \frac{I\{A = d(X_i; \beta)\}}{\hat{\rho}(A_i \mid X_i)\rho(A_i \mid X_i)} \right] \{Y - \hat{\mu}_d(X_i; \beta)\} \right) \\
=& \frac{1}{n} \sum_{i=1}^{n} \left( (2A_i - 1)\{e(X_i) - \hat{e}(X_i)\} \left[ \frac{I\{A = d(X_i; \beta)\}}{\hat{\rho}(A_i \mid X_i)\rho(A_i \mid X_i)} \right] \{Y - \mu_d(X_i; \beta)\} \right) \\
&+ \frac{1}{n} \sum_{i=1}^{n} \left( (2A_i - 1)\{e(X_i) - \hat{e}(X_i)\} \left[ \frac{I\{A = d(X_i; \beta)\}}{\hat{\rho}(A_i \mid X_i)\rho(A_i \mid X_i)} \right] \{\mu_d(X_i; \beta) - \hat{\mu}_d(X_i; \beta)\} \right).
\end{aligned}
$$

Since $\mathbb{E}[I\{A_i = d(X_i; \beta)\} \{Y - \mu_d(X_i; \beta)\}] = 0$ and Assumption 7, $\hat{V}_n(\beta) = \bar{v}_n(\beta) + o_p(n^{-1/2})$. Similarly, $\bar{v}_n(\beta) = V_n(\beta) + o_p(n^{-1/2})$ and therefore $\hat{V}_n(\beta) = V_n(\beta) + o_p(n^{-1/2})$.

Then we prove $\|\hat{\beta} - \beta_0\| = O_p(n^{-1/3})$. First, by Argmax Theorem, we have $\hat{\beta} \xrightarrow{p} \beta$. Next we apply Theorem 14.4 in Kosorok [2008] to show the converge rate. Take the Taylor expansion of $V(\beta)$ at $\beta = \beta_0$,

$$V(\beta) - V(\beta_0) = \frac{1}{2} \frac{\partial^2 V(\beta)}{\partial\beta\partial\beta^{\mathrm{T}}} \mid_{\beta=\beta_0} \|\beta - \beta_0\|^2 + o(\|\beta - \beta_0\|^2).$$

Since $\partial^2 V(\beta)/\partial\beta\partial\beta^{\mathrm{T}} < 0$, there exists $c_0 > 0$ such that $V(\beta) - V(\beta_0) < -c_0\|\beta - \beta_0\|^2$. Condition (i) holds.

For a sufficient small $R$,

$$\mathbb{E}\left[n^{1/2}\sup_{\|\beta-\beta_0\|\leq R}|\hat{V}_n(\beta) - V(\beta) - \{\hat{V}_n(\beta_0) - V(\beta_0)\}|\right]$$

$$=\mathbb{E}\left[n^{1/2}\sup_{\|\beta-\beta_0\|\leq R}|\hat{V}_n(\beta) - V_n(\beta) + V_n(\beta) - V(\beta) - \{\hat{V}_n(\beta_0) - V_n(\beta_0) + V_n(\beta_0) - V(\beta_0)\}|\right]$$

$$\leq\mathbb{E}\left[n^{1/2}\sup_{\|\beta-\beta_0\|\leq R}|\hat{V}_n(\beta) - V_n(\beta) - \{\hat{V}_n(\beta_0) - V_n(\beta_0)\}|\right]$$

$$+\mathbb{E}\left[n^{1/2}\sup_{\|\|\beta-\beta_0\|\leq R}|V_n(\beta) - V(\beta) - \{V_n(\beta_0) - V(\beta_0)\}|\right]$$

$$:\gamma_1 + \gamma_2.$$

Because $\hat{V}_n(\beta) = V_n(\beta) + o_p(n^{-1/2})$, $\gamma_1 = o_p(1)$. Further,

$$V_n(\beta) - V_n(\beta_0)$$

$$= \frac{1}{n}\sum_{i=1}^n\left[\frac{I\{A_i = d(X_i;\beta)\}}{\rho(A_i \mid X_i)}\{Y_i - \mu_d(X_i;\beta)\} + \mu_d(X_i;\beta)\right.$$

$$\left.- \frac{I\{A_i = d(X_i;\beta_0)\}}{\rho(A_i \mid X_i)}\{Y_i - \mu_d(X_i;\beta_0)\} - \mu_d(X_i;\beta_0)\right]$$

$$= \frac{1}{n}\sum_{i=1}^n\left\{\frac{(2A_i - 1)Y_i - \mu_1(X_i)A_i + \mu_0(X_i)(1 - A_i)}{\rho(A_i \mid X_i)} + \mu_1(X_i) - \mu_0(X_i)\right\}$$

$$\times \left\{I(X_i^{\mathrm{T}}\beta > 0) - I(X_i^{\mathrm{T}}\beta_0 > 0)\right\}.$$

Denote $G_R(\cdot)$ as the envelope of the class

$$\mathcal{F}_\beta(y, a, x) = \left[\left\{\frac{(2a - 1)y - \mu_1(x)a + \mu_0(x)(1 - a)}{\rho(a \mid x)} + \mu_1(x) - \mu_0(x)\right\}\right.$$

$$\left.\times \left\{I(x^{\mathrm{T}}\beta > 0) - I(x^{\mathrm{T}}\beta_0 > 0)\right\} : \|\beta - \beta_0\| < R\right].$$

Define $M$ as

$$M = \sup|\frac{(2a - 1)y - \mu_1(x)a + \mu_0(x)(1 - a)}{\rho(a \mid x)} + \mu_1(x) - \mu_0(x)|.$$

By Assumptions 3, 4 and 5, $M < \infty$. Because $X$ is bounded, there exists a constant $0 < k_0 < \infty$ s.t. $|x^{\mathrm{T}}\beta - x^{\mathrm{T}}\beta_0| < k_0R$ when $\|\beta - \beta_0\|_2 < R$. For the indicator function $I(-k_0R \leq x^{\mathrm{T}}\beta_0 \leq k_0R)$,

1. when $-k_0R \leq x^{\mathrm{T}}\beta_0 \leq k_0R, I(-k_0R \leq x^{\mathrm{T}}\beta_0 \leq k_0R) = 1 \geq |I(x^{\mathrm{T}}\beta > 0) - I(x^{\mathrm{T}}\beta_0 > 0)|$.
2. when $x^{\mathrm{T}}\beta_0 > k_0R, x^{\mathrm{T}}\beta = x^{\mathrm{T}}(\beta - \beta_0) + x^{\mathrm{T}}\beta_0 > -k_0R + k_0R = 0, I(-k_0R \leq x^{\mathrm{T}}\beta_0 \leq k_0R) = 0 = |I(x^{\mathrm{T}}\beta > 0) - I(x^{\mathrm{T}}\beta_0 > 0)|$.
3. when $x^{\mathrm{T}}\beta_0 < -k_0R, x^{\mathrm{T}}\beta = x^{\mathrm{T}}(\beta - \beta_0) + x^{\mathrm{T}}\beta_0 < k_0R + (-k_0R) = 0, I(-k_0R \leq x^{\mathrm{T}}\beta_0 \leq k_0R) = 0 = |I(x^{\mathrm{T}}\beta > 0) - I(x^{\mathrm{T}}\beta_0 > 0)|$.

Hence, define $G_R(\cdot) = MI(-k_0R \leq x^{\mathrm{T}}\beta_0 \leq k_0R)$. By Assumption 6, there exists a positive constant $k_1$ such that $\mathbb{E}G_R^2 = M^2Pr(-k_0R \leq x^{\mathrm{T}}R \leq k_0R) \leq M^2(k_12k_0R) < \infty$. Because $\mathcal{F}_\beta$ is a class of indicate functions, $\mathcal{F}_\beta$ is a VC

class of functions and its entropy, denoted as $\mathcal{J}(\mathcal{F})$, is finite. Consider the empirical process

$$\mathbb{G}_n \mathcal{F}_\beta = n^{-1/2} \sum_{i=1}^{n} [\mathcal{F}_\beta(Y_i, A_i, X_i) - \mathbb{E}\{\mathcal{F}_\beta(Y_i, A_i, X_i)\}]$$
$$= n^{1/2} [V_n(\beta) - V_n(\beta_0) - \{V(\beta) - V(\beta_0)\}]$$
$$= n^{1/2} [V_n(\beta) - V(\beta) - \{V_n(\beta_0) - V(\beta_0)\}],$$

we have

$$\gamma_2 = \mathbb{E}\left[ n^{1/2} \sup_{\|\|\beta - \beta_0\| \le R} |V_n(\beta) - V(\beta) - \{V_n(\beta_0) - V(\beta_0)\}| \right]$$
$$= \mathbb{E}\left[ n^{1/2} \sup_{\|\|\beta - \beta_0\| \le R} |\mathbb{G}_n \mathcal{F}_\beta| \right] \le c_1 \mathcal{J}(\mathcal{F}) \sqrt{\mathbb{E} G_R^2} = c_1 \mathcal{J}(\mathcal{F}) M \sqrt{2 k_1 k_0} R^{1/2} \le C_1 R^{1/2},$$

where $c_1$ is a finite constant and $C_1 = c_1 \mathcal{J}(\mathcal{F}) M \sqrt{2 k_1 k_0} < \infty$. Therefore

$$\mathbb{E}\left[ n^{1/2} \sup_{\|\beta - \beta_0\| \le R} |\hat{V}_n(\beta) - V(\beta) - \{\hat{V}_n(\beta_0) - V(\beta_0)\}| \right] \le C_1 R^{1/2}.$$

Let $\phi_n(R) = R^{1/2}$ and $\alpha = 3/2 < 2$. Then $\phi_n(R)/R^\alpha = R^{-1}$ is decreasing and not depend on $n$. Condition (ii) holds.

Let $r_n = n^{1/3}$, then $r_n^2 \phi_n(r_n^{-1}) = n^{2/3} n^{-1/6} = n^{1/2}$. Condition (iii) holds. Because $\hat{V}_n(\hat{\beta}) \ge \sup_\beta \hat{V}_n(\beta)$, we have $\|\hat{\beta} - \beta_0\| = O_p(n^{-1/3})$.

Finally, we derive the asymptotic distribution of $\hat{V}_n(\hat{\beta})$.

$$\sqrt{n}\left\{ \hat{V}_n(\hat{\beta}) - V(\beta_0) \right\}$$
$$= \sqrt{n}\left\{ \hat{V}_n(\hat{\beta}) - \hat{V}_n(\beta_0) + \hat{V}_n(\beta_0) - V(\beta_0) \right\}.$$

We first consider $\sqrt{n}\left\{ \hat{V}_n(\hat{\beta}) - \hat{V}_n(\beta_0) \right\}$, which can be decomposed as

$\sqrt{n}\left\{ \hat{V}_n(\hat{\beta}) - \hat{V}_n(\beta_0) \right\} = \sqrt{n}\left[ \hat{V}_n(\hat{\beta}) - \hat{V}_n(\beta_0) - \left\{ V(\hat{\beta}) - V(\beta_0) \right\} \right] + \sqrt{n}\left\{ V(\hat{\beta}) - V(\beta_0) \right\}$. Take the Taylor expansion of $V(\beta)$ at $\beta = \beta_0$,

$$\sqrt{n}\left\{ V(\hat{\beta}) - V(\beta_0) \right\} = \sqrt{n}\left\{ \frac{1}{2} \frac{\partial^2 V(\beta)}{\partial \beta \partial \beta^{\mathrm{T}}} |_{\beta = \beta_0} \|\hat{\beta} - \beta_0\|^2 + o(\|\hat{\beta} - \beta_0\|^2) \right\}$$
$$= O_p(n^{-1/6}) = o_p(1).$$

And

$$\sqrt{n}\left[ \hat{V}_n(\hat{\beta}) - \hat{V}_n(\beta_0) - \left\{ V(\hat{\beta}) - V(\beta_0) \right\} \right]$$
$$\le \mathbb{E}\left[ n^{1/2} \sup_{\|\beta - \beta_0\| \le c_2 n^{-1/3}} |\hat{V}_n(\beta) - V(\beta) - \{\hat{V}_n(\beta_0) - V(\beta_0)\}| \right]$$
$$\le C_1 \sqrt{c_2 n^{-1/3}} = o_p(1),$$

where $c_2$ is a constant such that $\|\hat{\beta} - \beta_0\| = c_2 n^{1/3}$. Therefore $\sqrt{n}\left\{ \hat{V}_n(\hat{\beta}) - \hat{V}_n(\beta_0) \right\} = o_p(1)$. Next we consider $\sqrt{n}\left\{ \hat{V}_n(\beta_0) - V(\beta_0) \right\}$.

$$\sqrt{n}\left\{ \hat{V}_n(\beta_0) - V(\beta_0) \right\} = \sqrt{n}\left\{ \hat{V}_n(\beta_0) - V_n(\beta_0) + V_n(\beta_0) - V(\beta_0) \right\}$$
$$= o_p(1) + \sqrt{n}\left\{ V_n(\beta_0) - V(\beta_0) \right\}.$$

$$\sqrt{n}\left\{V_n(\beta_0) - V(\beta_0)\right\} = \frac{1}{\sqrt{n}}\sum_{i=1}^{n}\left[\frac{I\left\{A_i = d(X_i;\beta_0)\right\}}{\rho(A_i \mid X_i)}\left\{Y_i - \mu_d(X_i;\beta_0)\right\} + \mu_d(X_i;\beta_0) - V(\beta_0)\right]$$

$$= \frac{1}{\sqrt{n}}\sum_{i=1}^{n}\varepsilon_i \xrightarrow{D} \mathcal{N}(0,\sigma^2).$$

where $\varepsilon_i = \frac{I\{A_i = d(X_i;\beta_0)\}}{\rho(A_i \mid X_i)}\left\{Y_i - \mu_d(X_i;\beta_0)\right\} + \mu_d(X_i;\beta_0) - V(\beta_0)$, and $\sigma^2 = \mathbb{E}\left(\varepsilon_i^2\right)$. Therefore,

$$\sqrt{n}\left\{\hat{V}_n(\hat{\beta}) - V(\beta_0)\right\} = \sqrt{n}\left\{\hat{V}_n(\hat{\beta}) - \hat{V}_n(\beta_0) + \hat{V}_n(\beta_0) - V_n(\beta_0) + V_n(\beta_0) - V(\beta_0)\right\}$$

$$= o_p(1) + o_p(1) + \sqrt{n}\left\{V_n(\beta_0) - V(\beta_0)\right\}$$

$$\xrightarrow{D} \mathcal{N}(0,\sigma^2).$$

# B  PROOF OF THEOREM 2

To derive the asymptotic distribution of $\hat{\beta}$, we adhere to the main theorem in Kim and Pollard [1990]. We begin to verify the assumptions laid down in Kim and Pollard [1990] 's main theorem. First define

$$g(\cdot,\beta) = \frac{I\left\{A = d(X;\beta)\right\}}{\rho(A \mid X)}\left\{Y - \mu_d(X;\beta)\right\} + \mu_d(X;\beta)$$

$$- \frac{I\left\{A = d(X;\beta_0)\right\}}{\rho(A \mid X)}\left\{Y - \mu_d(X;\beta_0)\right\} - \mu_d(X;\beta_0)$$

$$\hat{g}(\cdot,\beta) = \frac{I\left\{A = d(X;\beta)\right\}}{\hat{\rho}(A \mid X)}\left\{Y - \hat{\mu}_d(X;\beta)\right\} + \hat{\mu}_d(X;\beta)$$

$$- \frac{I\left\{A = d(X;\beta_0)\right\}}{\hat{\rho}(A \mid X)}\left\{Y - \hat{\mu}_d(X;\beta_0)\right\} - \hat{\mu}_d(X;\beta_0)$$

where $g(\cdot,\beta_0) = \hat{g}(\cdot,\beta_0) = 0$. Notice that $P_n\hat{g}(\cdot,\beta) = \hat{V}_n(\beta) - \hat{V}_n(\beta_0)$ and $Pg(\cdot,\beta) = V(\beta) - V(\beta_0)$, therefore $\beta_0 = \operatorname{argmax}_\beta V(\beta) = \operatorname{argmax}_\beta Pg(\cdot,\beta)$ and $\hat{\beta} = \operatorname{argmax}_\beta P_n\hat{g}(\cdot,\beta)$.

Because

$$g(\cdot,\beta) = \frac{I\left\{A = d(X;\beta)\right\}}{\rho(A \mid X)}\left\{Y - \mu_d(X;\beta)\right\} + \mu_d(X;\beta)$$

$$- \frac{I\left\{A = d(X;\beta_0)\right\}}{\rho(A \mid X)}\left\{Y - \mu_d(X;\beta_0)\right\} - \mu_d(X;\beta_0)$$

$$= \left\{\frac{(2A-1)Y - \mu_1(X)A + \mu_0(X)(1-A)}{\rho(A \mid X)} + \mu_1(X) - \mu_0(X)\right\}$$

$$\times \left\{I(X^{\mathrm{T}}\beta > 0) - I(X^{\mathrm{T}}\beta_0 > 0)\right\}.$$

Then denote $G_R(\cdot)$ as the envelope of class

$$\mathcal{F}_\beta(y,a,x) = \left[\left\{\frac{(2a-1)y - \mu_1(x)a + \mu_0(x)(1-a)}{\rho(a \mid x)} + \mu_1(x) - \mu_0(x)\right\}\right.$$

$$\left. \times \left\{I(x^{\mathrm{T}}\beta > 0) - I(x^{\mathrm{T}}\beta_0 > 0)\right\} : \|\beta - \beta_0\| < R\right].$$

From Theorem 1, $PG_R^2 = O_p(R)$, condition (vi) holds. Similarly, condition (vii) in Kim and Pollard [1990] can be verified. Given that $\mathcal{F}_\beta(y,a,x)$ is a class of indicate functions, it follows that $G_R(\cdot)$ complies with the uniform manageability condition.

Next it is shown

$$\sup_\beta P_ng(\cdot,\beta) - P_ng(\cdot,\hat{\beta}) \leq o_p(n^{-2/3}).$$

Similar to the proof of Lemma 2 in Wang et al. [2018], denote

$$\tilde{g}(\cdot, \beta, \sigma, \gamma) = \left[\frac{(2A-1)Y - \{\mu_1(X) + \gamma\}A + \{\mu_0(X) + \gamma\}(1-A)}{\{e(X) + \sigma\}A + (1-A)\{1 - e(X) - \sigma\}} + \mu_1(X) - \mu_0(X)\right]$$
$$\times \left\{I(X^{\mathrm{T}}\beta > 0) - I(X^{\mathrm{T}}\beta_0 > 0)\right\}$$

And $\beta_0 = \mathrm{argmax} P\tilde{g}(\cdot, \beta, 0, 0)$. Define

$$W_n(t, \sigma, \gamma) = n^{2/3}(P_n - P)\tilde{g}(\cdot, \beta_0 + tn^{-1/3}, \sigma n^{-1/3}, \gamma n^{-1/3}).$$

Given that $\sup_{x \in \mathcal{X}} |\hat{e}(X) - e(X)| = o_p(n^{-1/3})$ and $\sup_{x \in X} |\mu_a(X) - \hat{\mu}_a(X)| = o_p(n^{-1/3})$ for $a = 1, 0$, it follows that

$$W_n\left[n^{1/3}(\beta - \beta_0), n^{1/3}\{\hat{e}(X) - e(X)\}, n^{1/3}\{\hat{\mu}_A(X) - \mu_A(X)\}\right] - W_n\left[n^{1/3}(\beta - \beta_0), 0, 0\right] = o_p(1),$$
$$(P_n - P)\left[\tilde{g}\{\cdot, \beta, \hat{e}(X) - e(X), \hat{\mu}_A(X) - \mu_A(X)\} - \tilde{g}(\cdot, \beta, 0, 0)\right] = o_p(n^{-2/3}),$$
$$(P_n - P)\{\hat{g}(\cdot, \beta) - g(\cdot, \beta)\} = o_p(n^{-2/3}),$$
$$P_n\hat{g}(\cdot, \beta) - P_n g(\cdot, \beta) - P\hat{g}(\cdot, \beta) + Pg(\cdot, \beta) = o_p(n^{-2/3}).$$

Take the Taylor expansion of $\tilde{g}(\cdot, \beta, \sigma, \gamma)$ and $\tilde{g}(\cdot, \beta, 0, 0)$ at $(\beta_0, 0, 0)$, it is found that $Pg(\cdot, \beta) - P\hat{g}(\cdot, \beta) = P\tilde{g}(\cdot, \beta, 0, 0) - P\tilde{g}(\cdot, \beta, \hat{e}(X) - e(X), \hat{\mu}_A(X) - \mu_A(X)) = o_p(n^{-2/3})$. Therefore, for any $\beta$ uniformly in a $O(n^{-1/3})$ neighborhood of $\beta_0$, it holds that $P_n\hat{g}(\cdot, \beta) - P_n g(\cdot, \beta) = o_p(n^{-2/3})$. Define $\tilde{\beta} = \mathrm{argmax}_\beta P_n g(\cdot, \beta)$, it follows that

$$P_n g(\cdot, \hat{\beta}) = P_n\hat{g}(\cdot, \hat{\beta}) - o_p(n^{-2/3}) \geq P_n\hat{g}(\cdot, \tilde{\beta}) - o_p(n^{-2/3}) = P_n g(\cdot, \tilde{\beta}) - o_p(n^{-2/3}).$$

Condition (i) holds. Lastly, we verify conditions (iv) and (v). First we calculate

$$H = -\frac{\partial^2 Pg(\cdot, \beta)}{\partial\beta\partial\beta^{\mathrm{T}}}\Big|_{\beta=\beta_0}.$$

$$Pg(\beta) = \mathbb{E}\left[\frac{I\{A = d(X; \beta)\}}{\rho(A \mid X)}\{Y - \mu_d(X; \beta)\} + \mu_d(X; \beta)\right.$$
$$\left. - \frac{I\{A = d(X; \beta_0)\}}{\rho(A \mid X)}\{Y - \mu_d(X; \beta_0)\} - \mu_d(X; \beta_0)\right]$$
$$= \mathbb{E}\left[Y(1)d(X; \beta) + Y(0)\{1 - d(X; \beta)\}\right] - \mathbb{E}\left[Y(1)d(X; \beta_0) + Y(0)\{1 - d(X; \beta_0)\}\right]$$
$$= \mathbb{E}\left[\{Y(1) - Y(0)\}\{I(X^{\mathrm{T}}\beta > 0) - I(X^{\mathrm{T}}\beta_0 > 0)\}\right]$$
$$= \mathbb{E}\left(\{I(X^{\mathrm{T}}\beta > 0) - I(X^{\mathrm{T}}\beta_0 > 0)\}\mathbb{E}\{Y(1) - Y(0) \mid X\}\right)$$
$$= \mathbb{E}\left[\{I(X^{\mathrm{T}}\beta > 0) - I(X^{\mathrm{T}}\beta_0 > 0)\}h(X)\right],$$

where $h(X) = \mathbb{E}\{Y(1) - Y(0) \mid X\}$. Then

$$\frac{\partial Pg(\beta)}{\partial\beta} = \frac{\partial\mathbb{E}\left[\{I(X^{\mathrm{T}}\beta > 0) - I(X^{\mathrm{T}}\beta_0 > 0)\}h(X)\right]}{\partial\beta}.$$

Similarly in Example 6.4 in Kim and Pollard [1990] and the proof of Theorem 1 in Wang et al. [2018], denote $T_\beta = (I - \|\beta\|^{-2}\beta\beta^{\mathrm{T}})(I - \beta_0\beta_0^{\mathrm{T}}) + \|\beta\|^{-1}\beta\beta_0^{\mathrm{T}}$, where I is the identity matrix, maps $A = \{x^{\mathrm{T}}\beta_0 > 0\}$ onto $A(\beta) = \{x^{\mathrm{T}}\beta > 0\}$, taking $\partial A$ onto $\partial A(\beta)$. The surface measure $\sigma_\beta$ on $\partial A(\beta)$ has the constant density $p_\beta = \beta^{\mathrm{T}}\beta_0/\|\beta\|$ with respect to the image of the surface measure $\sigma = \sigma_{\beta_0}$ under $T_\beta$. The outward pointing normal to $A(\beta)$ is the standardized vector $-\beta/\|\beta\|$ and along $\partial A$ the derivative $(\partial/\partial\beta)T_\beta x$ reduces to $-\|\beta\|^{-2}\{\beta x^{\mathrm{T}} + (\beta^{\mathrm{T}}x)I\}$.

Then

$$\frac{\partial\mathbb{E}\left[\{I(X^{\mathrm{T}}\beta > 0) - I(X^{\mathrm{T}}\beta_0 > 0)\}h(X)\right]}{\partial\beta}$$
$$= \|\beta\|^{-2}\beta^{\mathrm{T}}\beta_0(I + \|\beta\|^{-2}\beta\beta^{\mathrm{T}})\int\{x^{\mathrm{T}}\beta_0 = 0\}f(T_\beta x)h(T_\beta x)x d\sigma.$$

Given that $T_{\beta_0} x = x$ along $\{x^T \beta_0 = 0\}$ and

$$\frac{\partial \mathbb{E}\left[\left\{I(X^T \beta > 0) - I(X^T \beta_0 > 0)\right\} h(X)\right]}{\partial \beta} \Big|_{\beta = \beta_0} = 0,$$

it is follows that $\int \{x^T \beta_0 = 0\} f(x) h(x) x d\sigma = 0$. Using the fact $\|\beta_0\| = 1$,

$$\begin{aligned}
-H &= \frac{\partial^2 Pg(\cdot, \beta)}{\partial \beta \partial \beta^T} \Big|_{\beta = \beta_0} = \frac{\partial \|\beta\|^{-2} \beta^T \beta_0 (I + \|\beta\|^{-2} \beta \beta^T)}{\partial \beta^T} \Big|_{\beta = \beta_0} \int \{x^T \beta_0 = 0\} f(x) h(x) x d\sigma \\
&\quad + (I + \|\beta_0\|^{-2} \beta_0 \beta_0^T) \frac{\partial \int \{x^T \beta_0 = 0\} f(T_\beta x) h(T_\beta x) x d\sigma}{\partial \beta^T} \Big|_{\beta = \beta_0} \\
&= -(I + \beta_0 \beta_0^T) \int \{x^T \beta_0 = 0\} \left\{ \dot{f}(x) h(x) + f(x) \dot{h}(x) \right\}^T \beta_0 x x^T d\sigma \\
&= -\int \{x^T \beta_0 = 0\} \left\{ \dot{f}(x) h(x) + f(x) \dot{h}(x) \right\}^T \beta_0 x x^T d\sigma,
\end{aligned}$$

where $\dot{f}(x)$ and $\dot{h}(x)$ denote the first-gradient with respect to $x$.

Next, we derive the covariance kernel function $C(s,t) = \lim_{\alpha \to \infty} \alpha \mathbb{E} g(\cdot, \beta_0 + s/\alpha) g(\cdot, \beta_0 + t/\alpha)$. Because $2\mathbb{E} g(\cdot, \beta_0 + s/\alpha) g(\cdot, \beta_0 + t/\alpha) = \mathbb{E}|g(\cdot, \beta_0 + s/\alpha) - g(\cdot, \beta_0)|^2 + \mathbb{E}|g(\cdot, \beta_0 + t/\alpha) - g(\cdot, \beta_0)|^2 - \mathbb{E}|g(\cdot, \beta_0 + s/\alpha) - g(\cdot, \beta_0 + t/\alpha)|^2$, it is necessary only to calculate $\mathbb{E}|g(\cdot, \beta_0 + s/\alpha) - g(\cdot, \beta_0 + t/\alpha)|^2$. It follows that

$$\begin{aligned}
&|g(\cdot, \beta_0 + s/\alpha) - g(\cdot, \beta_0 + t/\alpha)|^2 \\
&= \left\{ \frac{(2A - 1)Y - \mu_1(X)A + \mu_0(X)(1 - A)}{\rho(A \mid X)} + \mu_1(X) - \mu_0(X) \right\}^2 \\
&\quad \times |I(X^T(\beta_0 + \frac{s}{\alpha}) > 0) - I(X^T(\beta_0 + \frac{t}{\alpha}) > 0)|.
\end{aligned}$$

And

$$\begin{aligned}
&\alpha \mathbb{E}|g(\cdot, \beta_0 + \frac{s}{\alpha}) - g(\cdot, \beta_0 + \frac{t}{\alpha})|^2 \\
&= \alpha \mathbb{E}\left\{ |I(X^T(\beta_0 + \frac{s}{\alpha}) > 0) - I(X^T(\beta_0 + \frac{t}{\alpha}) > 0)| \times S_X \right\},
\end{aligned}$$

where

$$S_X = \mathbb{E}\left[ \left\{ \frac{(2A - 1)Y - \mu_1(X)A + \mu_0(X)(1 - A)}{\rho(A \mid X)} + \mu_1(X) - \mu_0(X) \right\}^2 \mid X \right].$$

Similarly in Example 6.4 in Kim and Pollard [1990] and Wang et al. [2018], define $\beta(\tau) = \sqrt{1 - \|\tau\|^2} \beta_0 + \tau$, where $\tau$ is orthogonal to $\beta_0$ and ranges over a neighborhood of the origin. Given the fact that the parameter space is on the sphere ($\|\beta\| = 1, \|\beta_0\| = 1$), such a decomposition can be obtained by taking $\tau = \tau(\beta) = T_0 \beta$, where $T_0 = I - \beta_0 \beta_0^T$. Then $\beta = (\beta_0^T \beta) \beta_0 + T_0 \beta$ such that $\beta_0^T \beta = \sqrt{1 - \|\tau\|^2}$ and $\beta_0^T \tau = \beta_0^T T_0 \beta = 0$. Then we have $\tau(\beta_0 + s/\alpha) = T_0 s/\alpha, \tau(\beta_0 + t/\alpha) = T_0 t/\alpha$. Similarly, we can decompose $X$ as $X = r\beta_0 + Z$ with a random variable $r$ and a random vector $Z$, where $Z$ is orthogonal to $\beta_0$. Denote $s^* = T_0 s$ and $t^* = T_0 t$, then it follows

$$\begin{aligned}
X^T(\beta_0 + \frac{t}{\alpha}) &= (r\beta_0 + Z)^T(\sqrt{1 - \|\tau(\beta_0 + \frac{t}{\alpha})\|^2} \beta_0 + \tau(\beta_0 + \frac{t}{\alpha})) \\
&= (r\beta_0 + Z)^T(\sqrt{1 - \frac{\|t^*\|^2}{\alpha^2}} \beta_0 + T_0 \frac{t}{\alpha}) \\
&= r\sqrt{1 - \frac{\|t^*\|^2}{\alpha^2}} + Z^T \frac{t^*}{\alpha}.
\end{aligned}$$

Table 4: Estimates for the linear regime (denoted as "est"), the corresponding 95% confidence intervals (denoted as "CI") and the confidence interval lengths (denoted as "Length") when $\epsilon_n = 0.7$.

| $\epsilon_n = 0.7$ | Int | age | BMI | Temp | Glucose |
|---|---|---|---|---|---|
| est | 0.489 | 0.254 | 0.087 | 0.424 | -0.382 |
| CI | (-0.235, 0.719) | (-0.367, 0.452) | (-0.403, 0.577) | (-0.208, 0.613) | (-0.495, 0.097) |
| Length | 0.954 | 0.819 | 0.980 | 0.821 | 0.592 |
| | BUN | creatinine | WBC | bilirubin | BP |
| est | -0.279 | -0.162 | 0.486 | 0.133 | 0.072 |
| CI | (-0.689, 0.339) | (-0.501, 0.428) | (-0.278, 0.813) | (-0.982, 0.991) | (-0.352, 0.391) |
| Length | 1.028 | 0.928 | 1.091 | 1.973 | 0.743 |

Define $p(\cdot, \cdot)$ as the joint probability distribution of $(r, Z)$. With a change of variable $w = \alpha r$, then $S_X = S_{r\beta_0 + Z} = S_{w\beta_0/\alpha + Z}$, we can rewrite $\alpha \mathbb{E}|g(\cdot, \beta_0 + s/\alpha) - g(\cdot, \beta_0 + t/\alpha)|^2$ as

$$\iint \left\{ -\frac{Z^{\mathrm{T}} s^*}{\sqrt{1 - \frac{\|s^*\|^2}{\alpha^2}}} > w \geq -\frac{Z^{\mathrm{T}} t^*}{\sqrt{1 - \frac{\|t^*\|^2}{\alpha^2}}} \right\} S_{\frac{w}{\alpha}\beta_0 + Z} p(\frac{w}{\alpha}, Z) dw dZ$$

$$+ \iint \left\{ -\frac{Z^{\mathrm{T}} t^*}{\sqrt{1 - \frac{\|t^*\|^2}{\alpha^2}}} > w \geq -\frac{Z^{\mathrm{T}} s^*}{\sqrt{1 - \frac{\|s^*\|^2}{\alpha^2}}} \right\} S_{\frac{w}{\alpha}\beta_0 + Z} p(\frac{w}{\alpha}, Z) dw dZ.$$

Integrate over $w$ and let $\alpha \to \infty$ to get

$$\lim_{\alpha \to \infty} \alpha \mathbb{E}|g(\cdot, \beta_0 + \frac{s}{\alpha}) - g(\cdot, \beta_0 + \frac{t}{\alpha})|^2 = \int |Z^{\mathrm{T}} s^* - Z^{\mathrm{T}} t^*| S_Z p(0, Z) dZ$$

$$= \int |Z^{\mathrm{T}} s - Z^{\mathrm{T}} t| S_Z p(0, Z) dZ$$

$$:= L(s - t),$$

with $L(s) \neq 0$ for $s \neq 0$. Therefore,

$$C(s, t) = \frac{L(s) + L(t) - L(s - t)}{2}.$$

.

# C  ADDITIONAL RESULTS FOR REAL DATA ANALYSIS

Table 4 presents the confidence intervals for the estimated linear regimes in the eICU-CRD datasets when $\epsilon_n = 0.7$. However, at this $\epsilon_n$ value, the length of the confidence intervals is broad. As a result, these intervals don't pinpoint any significant covariates, leading to no findings.