# OpenReview forum: "Inference for Optimal Linear Treatment Regimes in Personalized Decision-making"
_auai.org/UAI/2024/Conference — UAI 2024 oral_

### Official Review · Reviewer_b7va · 2024-03-10

**Q2-1 Originality-Novelty:** 3
**Q2-2 Correctness-Technical Quality:** 3
**Q2-5 Clarity Of Writing:** 4

**Q10 Ethical Concerns:**

No, the eICU dataset used is public.

**Q1 Summary And Contributions:**

The paper presents a method to identify optimal linear treatment regimes by optimizing the AIPW estimator. By focusing on linear regimes, the paper aims to simplify the estimation and to ease interpretation in clinical contexts. It details the asymptotic behavior of the linear regime estimator, noting a cube-root convergence rate to a non-normal distribution. A bootstrap method is proposed to overcome the shortcomings of traditional nonparametric bootstrap methods. The approach is tested through simulations and applied to ICU data.

**Q2-3 Extent To Which Claims Are Supported By Evidence:**

3: Good: the main claims are supported by convincing evidence (in the form of adequate experimental evaluation, proofs, (pseudo-)code, references, assumptions).

**Q2-4 Reproducibility:**

4: Excellent: key resources (e.g. proofs, code, data) are available and key details (e.g. proof sketches, experimental setup) are comprehensively described for competent researchers to confidently and easily reproduce the main results.

**Q3 Main Strengths:**

The proposed approach not only addresses the interpretability and applicability of linear treatment regimes in clinical settings but also fills a gap in existing literature where the focus has predominantly been on the asymptotic distributions of the value function rather than the treatment regimes themselves.

Another significant contribution is the characterization of the asymptotic distribution of the estimated linear regime. The paper proves that the parameter indexing the linear treatment regime converges at a cube-root rate to a non-normal limiting distribution. This finding is particularly notable because it advances the understanding of the statistical properties of estimators within the framework of optimal treatment regimes --- a topic that has not been extensively studied in this specific context before.

The paper's adaptation of the bootstrap method proposed by Cattaneo et al. (2020) to derive confidence intervals for the estimated linear regimes also represents a contribution. The standard nonparametric bootstrap fails to provide accurate inference for estimators that converge at this rate due to their non-normal limiting distributions.

**Q4 Main Weakness:**

The focus on static settings limits the generalizability and applicability of the methodology to dynamic settings, where treatments and covariates vary over time. This limitation is notable given the significant portion of the literature on optimal treatment regimes that addresses dynamic settings, such as Murphy  (2003). Addressing this limitation, the authors could discuss potential extensions of their methodology to dynamic treatment regimes or justify the focus on static settings by describing specific contexts where static regimes are particularly relevant or sufficient.

Another concern is the claim of using semi-parametric or non-parametric machine learning models for estimating functions related to treatment effects. The paper actually utilizes GAMs for the simulation, which, while incorporating non-parametric elements like smoothing splines, still includes parametric aspects, potentially conflicting with the stated focus on non-parametric approaches.

In the dataset, the average CB is negative and substantially high in its standard deviation. This is atypical, as CB in an ICU setting is expected to be positive on average. Given that the study uses \( Y = -|CB| \) as the outcome, where higher values indicate a more aggressive approach to fluid management to avoid fluid excess, the negative average suggests a systemic approach to fluid restriction across the patient cohort in the eICU-CRD dataset. This is at odds with other ICU data, such as the study by van Mourik et al., which found an average positive CB (5139 mL).

van Mourik N, Metske HA, Hofstra JJ, et al. Cumulative fluid balance predicts mortality and increases time on mechanical ventilation in ARDS patients: An observational cohort study. PLoS One. 2019;14(10):e0224563. Published 2019 Oct 30. doi:10.1371/journal.pone.0224563

**Q5 Detailed Comments To The Authors:**

Pg.1 - The papers cited as examples of applying optimal treatment regimes in different fields are concerned with estimating average treatment effects of an intervention - not estimating optimal treatment regimes. Gerber and Green (2000), for instance, is concerned with estimating the ATE of a particular canvassing program on turnout, rather than the determining which canvassing regime yields the best average turnout.

More suitable examples are found in biostatistics and precision medicine, e.g.

Young, Jessica G., et al. "Comparative effectiveness of dynamic treatment regimes: an application of the parametric g-formula." Statistics in biosciences 3 (2011): 119-143.

Zhang, Zhongheng, Bin Zheng, and Nan Liu. "Individualized fluid administration for critically ill patients with sepsis with an interpretable dynamic treatment regimen model." Scientific Reports 10.1 (2020): 17874.

Pg.1 - Another doubly-robust method worth mentioning is targeted minimum loss-based estimation (TMLE). Unlike AIPTW, TMLE does not aim to solve an estimating equation, but
instead uses a log-likelihood loss function to minimize bias, which permits the use of nonparametric (ML) methods for estimation.

References:

Chapter 6 in: Van der Laan, Mark J., and Sherri Rose. Targeted learning: causal inference for observational and experimental data. Vol. 4. New York: Springer, 2011.

Poulos, Jason, et al. "Targeted learning in observational studies with multi‐valued treatments: An evaluation of antipsychotic drug treatment safety." Statistics in Medicine (2024).

Pg. 2 - It is stated that linear treatment regimes are more accessible than tree-based ones, although tree-based methodologies are common in clinical research due to their interpretability.

Pg. 3. - It is stated that the “focus [is] on the use of semi-parametric or non-parametric machine learning models” for estimation of the propensity score and conditional mean outcome, yet in the simulation these parameters are estimated via GAMs, which are not fully nonparametric because they include parametric components in the form of the additive predictors.

Pg. 7 - Why is the average CB in Table 2 negative? Even in an ICU population, average CB would be positive. For instance, in a different ICU dataset the average CB is 5139 mL.

**Q9 Complying With Reviewing Instructions:**

Yes

---

> ### Author Rebuttal · Authors · 2024-04-07
>
> Thank you for pointing out the focus on static settings and its potential limitation regarding dynamic treatment scenarios. We acknowledge the significance of extending our methodology to dynamic settings, as highlighted by existing literature, including Murphy (2003). Utilizing proof techniques akin to those in Wang et al. (2018) could facilitate the extension of our current analysis to accommodate dynamic settings. This promising direction for further research will be explored and added to the discussion section of our final manuscript.
>
> Thank you for your insightful observation regarding the second point. It's important to highlight that while Generalized Additive Models (GAMs) are the primary example used in our text, our approach indeed accommodates a wider range of models. Specifically, any semi-parametric or non-parametric model that achieves the requisite convergence rates, such as local polynomial regression, is compatible with our methodology.
>
> We appreciate your attention to the details noted in your third comment. We apologize for the typo in Table 2. The values presented reflect the mean and standard deviation for $Y=-\vert CB\vert$, rather than for CB alone, which accounts for the negative mean value observed. This has been corrected in the revised version of our manuscript.
>
> We're truly grateful for your detailed feedback on each page of our text. Your insights have been instrumental in refining our work, allowing us to develop a more comprehensive and clearer manuscript.
>
> Reference:
>
> Wang, Lan, et al. "Quantile-optimal treatment regimes." Journal of the American Statistical Association 113.523 (2018): 1243-1254.
>
> Murphy, Susan A. "Optimal dynamic treatment regimes." Journal of the Royal Statistical Society Series B: Statistical Methodology 65.2 (2003): 331-355.

---

### Official Review · Reviewer_WJLL · 2024-03-14

**Q2-1 Originality-Novelty:** 3
**Q2-2 Correctness-Technical Quality:** 3
**Q2-5 Clarity Of Writing:** 4

**Q10 Ethical Concerns:**

No.

**Q1 Summary And Contributions:**

The authors present an analysis of the convergence properties (n^(1/3)) of a doubly-robust estimator for optimal linear dynamic treatment regimes, and they give a modified bootstrap procedure for inference.

Detailed proofs are provided, as is a simulation study and real data application example.

**Q2-3 Extent To Which Claims Are Supported By Evidence:**

3: Good: the main claims are supported by convincing evidence (in the form of adequate experimental evaluation, proofs, (pseudo-)code, references, assumptions).

**Q2-4 Reproducibility:**

4: Excellent: key resources (e.g. proofs, code, data) are available and key details (e.g. proof sketches, experimental setup) are comprehensively described for competent researchers to confidently and easily reproduce the main results.

**Q3 Main Strengths:**

I believe this is the first treatment of linear optimal 1-stage regime estimators in this way.

The combination of the theoretical description of convergence and the practical algorithm is very valuable.

The authors build on existing work (e.g. Cattaneo, Kim, so on) in a clear and useful way.

The explanation is excellent, especially for such a highly technical topic. I enjoyed reading this!

**Q4 Main Weakness:**

The proposed method has a dependence on an unknown parameter, \epsilon_n, and the output appears to be sensitive to this. The authors are up front about this, and suggest caution.

The sizes for MC samples and bootstrap replicates seems small to me (100, 400) relative to what seems more normal (like 1000) especially if you're going to compare coverage and length across scenarios. There is more variance in Table 1 than I would like to see.

No discussion of time complexity/runtime. (Possibly related to the low numbers of replicates.)

**Q5 Detailed Comments To The Authors:**

As I mentioned, I really enjoyed this paper. I have three main comments and a few minor ones.

1) I appreciate being forthright about the dependence on \epsilon_n, and the advice to maybe try some different ones. However, I think some things are not clear. For example, what is your opinion about whether the values identified in Table 1 apply in all/most circumstances? If you want somebody to actually use this, what do they write in their methods section? There is space in the current manuscript to expand on how sensitive you believe the bootstrap distribution to be to \epsilon_n, and to suggest what you think at this point would be "best practice." I note that the value 0.3 used in the example doesn't show up in the simulation, for example.

2) Especially if you are going to lean on simulation to get a sense of what \epsilon_n should be, the number of simulations currently used (only 100, with only 400 bootstraps)

3) There's no discussion of runtime. UAI is an interdisciplinary conference with a substantial computer science foundation; some discussion of computation is appropriate.

Minor comments:

"Given its cube-root convergence, a substantial number of observations is advantageous in real-world examples." - I don't understand what this means. Do you mean, "Its cube-root convergence means that larger datasets are required for use in real-world applications?" Try to be more direct in your language.

"... we facilitate the boostrap method..." - I think "facilitate" needs to be replaced by a more direct word.

"As an illustration, if we define..." I got confused here - maybe add "As an illustration, if we take l = 1 and define..."

Assumption 6: "...in an interior of B" - Earlier, B is defined to be the unit l-1 sphere so I'm not sure what this means.

Right before 3.1.2 there's a nice, concise statement of the main contributions of the paper. Consider (also?) having this near the end of your introduction.

"...under Assumption 10 is faster than that in Assumption 7..." - I know what you mean, but if somebody is reading fast they may just look at A7 and see o_p(n^(-1/2)) and get confused. Maybe refer to something like "the approach suggested following Assumption 7"?

Algorithms Step 4: extra "from"

"can either be refitted using the samples {X..." - Missing open parenthesis.

Table 1: "Monte Caro" - "Monte Carlo"

**Q9 Complying With Reviewing Instructions:**

Yes

---

> ### Author Rebuttal · Authors · 2024-04-07
>
> Thank you for your insightful comments and constructive criticism. In response to your first comment on the dependency on an unknown parameter $\epsilon_{n}$ and its sensitivity, we agree that identifying the optimal $\epsilon_{n}$ value , which minimizes the approximate Mean Squared Error, incorporates both the $H$ matrix and the covariance kernel $C(\cdot,\cdot)$, therefore the optimal value is case by case, and contingent on the specific data generation process at hand.
>
> In real-world applications, we suggest opting for $\epsilon_{n}$ values that correspond to local minima in confidence interval lengths. This approach informed our selection of 0.3 for the real data analysis, after testing $\epsilon_{n}$ values of {0.3, 0.5, 0.7}, among which {0.3, 0.5} were identified as local minima across most variables examined.
>
> Thanks for your concerns for the simulation sizes and bootstrap replicates, the current settings (100 for simulations and 400 for bootstraps) were constrained by available resources. However, recognizing the importance of robustness in our findings, we conducted an additional 100 simulations during the rebuttal period and plan to extend this further in the final version of our manuscript to enhance the statistical reliability of our results. Table 1 presents the estimates for $\beta_{01}$ and $\beta_{02}$ (denoted as “Est”) and their 95% quantile confidence interval length (“Length”) and coverage rate (“Coverage”) based on 200 simulations when $\epsilon_{n}=0.5$. These outcomes align closely with those observed from the initial 100 simulations, maintaining a coverage rate near the 95% benchmark.
>
> |       | Est   | Coverage | Length |
> |-------|-------|----------|--------|
> | β0₁   | 0.894 | 95.5%    | 0.086  |
> | β0₂   | 0.447 | 95.5%    | 0.175  |
>
> *Table 1: Simulation results when $\epsilon_n = 0.5$ based on 200 Monte Carlo times with 400 bootstrap samples in each simulation time.*
>
> We appreciate your recommendation regarding the inclusion of runtime analysis. The computational complexity is represented as $O(KB)$, where $B$ is the size of bootstrap samples, and $K$ denotes the algorithm's complexity for obtaining the estimate $\hat{\beta}$ given $\hat{V}_n(\beta)$. For instance, utilizing a genetic algorithm implies $K=O(GNn)$, with $G$ indicating the number of iterations and $N$ the population size. We have incorporated this discussion into the final version of our paper for a comprehensive analysis.
>
> Lastly, we are grateful for your detailed reading and the minor suggestions provided. We have meticulously reviewed and incorporated these changes into the final version of our paper, ensuring a more refined and insightful presentation of our research. Your feedback has been invaluable in enhancing the quality and clarity of our work, and we look forward to contributing meaningfully to the interdisciplinary discourse at the conference.

---

### Official Review · Reviewer_D4Xa · 2024-03-15

**Q2-1 Originality-Novelty:** 2
**Q2-2 Correctness-Technical Quality:** 3
**Q2-5 Clarity Of Writing:** 3

**Q1 Summary And Contributions:**

The paper considers the problem of learning optimal policies/ treatment regimes from observational data (off-policy learning). The main contribution is a characterization of the asymptotic distribution of the AIPTW-estimator for linear policies as well as the corresponding asymptotic distribution of the optimal policy. The authors then propose a bootstrap algorithm that can be used for uncertainty quantification in off-policy learning.

**Q2-3 Extent To Which Claims Are Supported By Evidence:**

3: Good: the main claims are supported by convincing evidence (in the form of adequate experimental evaluation, proofs, (pseudo-)code, references, assumptions).

**Q2-4 Reproducibility:**

3: Good: key resources (e.g. proofs, code, data) are available and key details (e.g. proofs, experimental setup) are sufficiently well-described for competent researchers to confidently reproduce the main results.

**Q3 Main Strengths:**

- Off-policy learning from observational data is of interest in many disciplines and uncertainty quantification methods are lacking
- The results are technically sound and validated empirically

**Q4 Main Weakness:**

- The results are limited to linear policies/ treatment regimes.
- The abstract and introduction could be written more concisely.

**Q5 Detailed Comments To The Authors:**

- Off-policy evaluation and learning are relevant in many disciplines beyond personalized medicine. I would suggest changing the title and the abstract/introduction to include other application domains as well (otherwise practitioners outside medicine might miss out on the results of the paper).

**Q9 Complying With Reviewing Instructions:**

Yes

---

> ### Author Rebuttal · Authors · 2024-04-07
>
> We appreciate your valuable feedback regarding the conciseness of our abstract and introduction, as well as the suggestion to broaden the scope of our title and abstract/introduction to encompass application domains beyond personalized medicine. Recognizing the relevance of off-policy evaluation and learning across various disciplines, we will revise our manuscript to highlight these broader applications.

---

### Official Review · Reviewer_rBpG · 2024-03-19

**Q2-1 Originality-Novelty:** 2
**Q2-2 Correctness-Technical Quality:** 3
**Q2-5 Clarity Of Writing:** 3

**Q1 Summary And Contributions:**

In this paper the authors presented the asymptotic distribution of the parameters of the optimal linear treatment regime for Augmented Inverse Probability Weighting estimator. This allows the identification of the important covariates for the optimal linear treatment regime. The authors also proposed a bootstrap algorithm for estimating the distribution of the parameters of the linear regimes.

**Q2-3 Extent To Which Claims Are Supported By Evidence:**

2: Fair: the main claims are somewhat supported by evidence (but the experimental evaluation may be weak, or does not match entirely with the claims, important baselines may be missing, proofs contain important ideas but lack rigor, algorithmic details are only discussed superficially, references are imprecise, assumptions are not sufficiently motivated or explicated, etc.).

**Q2-4 Reproducibility:**

3: Good: key resources (e.g. proofs, code, data) are available and key details (e.g. proofs, experimental setup) are sufficiently well-described for competent researchers to confidently reproduce the main results.

**Q3 Main Strengths:**

1. The asymptotic distribution of the parameters of the optimal linear treatment regime would allow the identification of the important covariates for the optimal linear treatment regime.
2. The proposed bootstrap algorithm could in practice offer an estimation the distribution of the parameters of the linear regimes.

**Q4 Main Weakness:**

1. The work is incremental.

**Q5 Detailed Comments To The Authors:**

While this paper presents interesting results about linear treatment regime, it is not clear why the authors would put "personalized medicine" in the title. Personalized medicine usually refers to treatment tailored to specific genetic variation of individual patients.

**Q9 Complying With Reviewing Instructions:**

Yes

---

> ### Author Rebuttal · Authors · 2024-04-07
>
> Acknowledging the feedback regarding the use of "personalized medicine" in our title, we have decided to revise it accordingly to more accurately reflect the scope of our study.

---

### Official Review · Reviewer_HrG6 · 2024-03-26

**Q2-1 Originality-Novelty:** 3
**Q2-2 Correctness-Technical Quality:** 3
**Q2-5 Clarity Of Writing:** 3

**Q1 Summary And Contributions:**

This paper establishes a series of statistical results for estimating the optimal linear treatment rule from observational data using a AIPW estimator for the expected outcome. Namely, they provide assumptions under which we have root-n and cube-root-n convergence for the estimated expected outcome and treatment rule coefficients, respectively. Additionally, they prove that, under some additional conditions, the coefficient estimates converge to a gaussian distribution. The significant benefit of these results is that they allow for the construction of confidence intervals around the coefficients of the linear treatment rule.

**Q2-3 Extent To Which Claims Are Supported By Evidence:**

3: Good: the main claims are supported by convincing evidence (in the form of adequate experimental evaluation, proofs, (pseudo-)code, references, assumptions).

**Q2-4 Reproducibility:**

4: Excellent: key resources (e.g. proofs, code, data) are available and key details (e.g. proof sketches, experimental setup) are comprehensively described for competent researchers to confidently and easily reproduce the main results.

**Q3 Main Strengths:**

Despite their importance, formal statistical guarantees are overlooked in much of the ML-for-causal-inference literature and I appreciate the author’s efforts in this area. I think the results are valuable, if potentially hard to apply in practice. While the results are highly technical and, in places, complex, I found the writing clear and reasonably easy to follow.

**Q4 Main Weakness:**

See below.

**Q5 Detailed Comments To The Authors:**

1. I think the authors did a reasonable job of giving intuition for many of the assumptions, but several, especially 2-9, still felt a bit under-explained and I’m not sure I would be able to assess whether they hold in a practical settings
2. I think it would be worth mentioning some of the other statistical work on optimal treatment rules. The work that comes to mind is that of Alex Luedtke. Though I am not aware of results for linear treatment rules, their work certainly provides statistical guarantees for the estimated value function under the estimated optimal treatment rule.
3. It raises a very large red flag for me that BP and/or mean arterial pressure (MAP) are not included in the features relevant to administering vasopressors. Vasopressors work by restricting blood vessels to raise blood pressure, so, while hypotension does not *need* to be present to administer vasopressors, it is certainly the most common indicator that they are needed.

**Q9 Complying With Reviewing Instructions:**

Yes

---

> ### Author Rebuttal · Authors · 2024-04-07
>
> Thank you for your constructive feedback on the explanations provided for assumptions 2-9. Allow us to address each assumption in detail:
>
> Assumption 2 (the Stable Unit Treatment Value Assumption): this assumption is crucial in ensuring that an individual's potential outcomes are unaffected by the treatment allocations or outcomes of others. Its applicability depends on the specific context of the study. For instance, in the fields of epidemiology and infectious diseases, this assumption often does not hold due to the interconnected nature of individual outcomes.
>
> Assumption 3: this assumption focuses on the need for the treatments of interest to be observable across all patient subgroups, to verify Assumption 3 in real-world contexts, one should analyze the dataset to confirm that every possible combination of patient covariates is observable across each treatment group.
>
> Assumptions 4 and 5: these assumptions relate to standard regularity conditions essential for achieving asymptotic convergence, conditions we consider to be consistently held in practical scenarios.
>
> Assumption 6: this assumption introduces an identifiability condition, asserting the uniqueness of the true targeted optimal regime, which, while challenging to verify in practice, can be validated through extensive consultation with medical experts.
>
> Assumption 7: this assumption necessitates the convergence rate in the $L_{2}$ norm, which can be achieved through certain existing semi-parametric methods, such as Generalized Additive Models (GAM).
>
> Assumptions 8 and 9: these assumptions are technical conditions for the evaluation of the first and second-order derivatives of the value function and the kernel covariance, and they are typically met in practical applications.
>
> Secondly, incorporating your recommendation to engage more deeply with existing literature, including the contributions by Alex Luedtke, will enhance our discussion, and we will specifically include this analysis within the body of our main text to offer a more thorough introduction for our study.
>
> Thirdly, regarding the exclusion of mean arterial pressure (MAP) data in the context of vasopressor administration is acknowledged. In our study utilizing the eICU-CRD dataset, only mean blood pressure data were available. Despite initial analysis incorporating this variable, the estimated impact of mean blood pressure on the outcomes was minimal, at 0.072, leading to its exclusion from significant covariates in our model. Conversely, we chose to include temperature and white blood cell count, both of which exhibit a positive impact in the linear regime. This selection is medically intuitive, as sepsis commonly induces fever (Schortgen, 2012) and elevates white blood cell count (Munford, 2006), aligning with findings from medical research.
>
> Reference:
>
> Schortgen, F. "Fever in sepsis." Minerva anestesiologica 78.11 (2012): 1254-1264.
>
> Munford, Robert S. "Severe sepsis and septic shock: the role of gram-negative bacteremia." Annu. Rev. Pathol. Mech. Dis. 1 (2006): 467-496.

---

### Meta-Review · Area_Chair_rzyA · 2024-04-18

The paper derives the asymptotic distribution for linear policies given the widely used AIPTW policy value estimator. Building on this result, they construct a consistent bootstrap estimator for the variance. Reviewers agreed the paper gives a strong and important new mathematical result that would be useful to the wider community of causal effect inference and off-policy evaluation from observational data. Reviewers made several useful suggestions which I trust the authors will incorporate into an updated version of the paper.